# Development and characterization of a chronic implant mouse model for vagus nerve stimulation

Ibrahim T Mughrabi[1†], Jordan Hickman[2,3†], Naveen Jayaprakash[1], Dane Thompson[1,4], Umair Ahmed[1], Eleni S Papadoyannis[5,6,7,8], Yao-Chuan Chang[1], Adam Abbas[1], Timir Datta-Chaudhuri[1], Eric H Chang[1], Theodoros P Zanos[1], Sunhee C Lee[9], Robert C Froemke[5,6,7,8], Kevin J Tracey[1], Cristin Welle[2,3*], Yousef Al-Abed[1], Stavros Zanos[1*]

[1]Institute of Bioelectronic Medicine, The Feinstein Institutes for Medical Research, Northwell Health, Manhasset, United States; [2]Departments of Neurosurgery, University of Colorado Anschutz Medical Campus, Aurora, United States; [3]Department of Physiology and Biophysics, University of Colorado Anschutz Medical Campus, Aurora, United States; [4]The Elmezzi Graduate School of Molecular Medicine, Manhasset, United States; [5]Skirball Institute for Biomolecular Medicine, New York University School of Medicine, New York University, New York, United States; [6]Department of Neuroscience and Physiology, Neuroscience Institute, Center for Neural Science, New York University School of Medicine, New York University, New York, United States; [7]Department of Otolaryngology, New York University School of Medicine, New York University, New York, United States; [8]Howard Hughes Medical Institute Faculty Scholar, New York University School of Medicine, New York University, New York, United States; [9]Institute of Molecular Medicine, The Feinstein Institutes for Medical Research, Northwell Health, Manhasset, United States

*For correspondence:
cristin.welle@cuanschutz.edu
(CW);
szanos@northwell.edu (SZ)

†These authors contributed
equally to this work

Competing interest: See
page 18

Reviewing editor: Isaac M Chiu,
Harvard Medical School, United
States

**Abstract** Vagus nerve stimulation (VNS) suppresses inflammation and autoimmune diseases in preclinical and clinical studies. The underlying molecular, neurological, and anatomical mechanisms have been well characterized using acute electrophysiological stimulation of the vagus. However, there are several unanswered mechanistic questions about the effects of chronic VNS, which require solving numerous technical challenges for a long-term interface with the vagus in mice. Here, we describe a scalable model for long-term VNS in mice developed and validated in four research laboratories. We observed significant heart rate responses for at least 4 weeks in 60–90% of animals. Device implantation did not impair vagus-mediated reflexes. VNS using this implant significantly suppressed TNF levels in endotoxemia. Histological examination of implanted nerves revealed fibrotic encapsulation without axonal pathology. This model may be useful to study the physiology of the vagus and provides a tool to systematically investigate long-term VNS as therapy for chronic diseases modeled in mice.

## Introduction

The vagus nerve (VN), the principal nerve of the parasympathetic nervous system, occupies a crucial role in the reflex control of physiological homeostasis (*Berthoud and Neuhuber, 2000*). The roles of VN reflexes in controlling the cardiovascular, pulmonary, and gastrointestinal systems have been studied for more than 100 years, but only recently has the role of VN reflexes been studied in

controlling inflammation (*Pavlov et al., 2020*; *Pavlov et al., 2018*; *Tracey, 2009*; *Tracey, 2002*). We and others have defined VN mechanisms, termed 'the inflammatory reflex (IR),' which inhibit cytokine production in the spleen, and attenuates inflammation in preclinical and clinical studies (*Borovikova et al., 2000*; *Rosas-Ballina et al., 2008*; *Rosas-Ballina et al., 2011*). Clinical VN stimulating devices are used in the treatment of epilepsy (*The Vagus Nerve Stimulation Study Group, 1995*) and depression (*Rush et al., 2000*), and are being studied in the treatment of brain disorders such as tinnitus (*Tyler et al., 2017*; *Vanneste et al., 2017*), stroke (*Engineer et al., 2019*; *Kimberley et al., 2018*), and Alzheimer's disease (*Merrill et al., 2006*; *Sjögren et al., 2002*), as well as peripheral organ and systemic diseases, such as heart failure (*De Ferrari et al., 2017*), cardiac arrhythmias (*Huang et al., 2015*; *Nasi-Er et al., 2019*; *Stavrakis et al., 2015*; *Yamaguchi et al., 2018*), pulmonary hypertension (*Ntiloudi et al., 2019*; *Yoshida et al., 2018*), rheumatoid arthritis (*Koopman et al., 2016*), Crohn's disease (*Bonaz et al., 2016*), and lupus (*Aranow, 2018*). Additional possible indications for vagus nerve stimulation (VNS) include common disorders in which inflammation is implicated, such as type 2 diabetes, obesity, and atherosclerosis (*Couzin-Frankel, 2010*; *Furman et al., 2019*; *Slavich, 2015*; *Strowig et al., 2012*).

Optimal preclinical studies of VNS in models of chronic disease require long-term implantation of a VNS device optimized for small animals, but to date the majority of chronically implanted VN devices have been limited to neurological and cardiovascular diseases in rats, pigs, dogs, and other large animals (*Nasi-Er et al., 2019*; *Yamaguchi et al., 2018*; *Yoshida et al., 2018*; *Annoni et al., 2019*; *Beaumont et al., 2016*; *Chinda et al., 2016*; *Farrand et al., 2017*; *Li et al., 2004*; *Meyers et al., 2018*; *Nuntaphum et al., 2018*; *Yamakawa et al., 2015*; *Ganzer et al., 2018*). The mouse is currently the species of choice in the study of disease pathophysiology, genetic mechanisms, and drug screening (*Bryda, 2013*; *Perlman, 2016*). However, VNS research in mice has been limited to acute stimulation (*Bansal et al., 2012*; *Caravaca et al., 2019*; *de Lucas-Cerrillo et al., 2011*; *Huffman et al., 2019*; *Huston et al., 2006*; *Meneses et al., 2016*; *Saeed et al., 2005*; *Shukla et al., 2015*; *The et al., 2007*), primarily because of significant surgical and technical challenges that accompany a mechanically (*Cuoco and Durand, 2000*; *Prasad et al., 2014*; *Rydevik et al., 1981*) and electrochemically (*McCreery et al., 1992*; *Negi et al., 2010*) stable long-term interface with the microscopic anatomy of the mouse VN. As a consequence, the therapeutic role of VNS in chronic inflammatory diseases and the long-term effects on neural mechanisms are largely unexplored. A functional and reliable long-term VNS implant in the mouse is necessary to broaden the translational potential of VNS in chronic diseases and provides an experimental tool to probe the long-term effects of VN neuromodulation on autonomic and neuroimmune circuits.

Here, we describe a surgical technique to permanently implant a micro-cuff electrode onto the mouse cervical vagus for long-term VNS—developed, refined, and validated through a collaboration between four research labs (two labs at Feinstein Institutes, University of Colorado, and New York University). This produced a standardized, long-lasting, functionally consistent implant with expected stimulus-elicited physiological responses that persist for at least 4 weeks. Because the implant does not interfere with physiological VN-mediated reflexes, including baroreflex, lung stretch reflex, and feeding reflexes, and successfully inhibits serum TNF levels in acute endotoxemia, this method may be useful in facilitating mechanistic studies of long-term VN neuromodulation.

## Materials and methods

### Key resources table

| Reagent type (species) or resource | Designation | Source or reference | Identifiers | Additional information |
|---|---|---|---|---|
| Commercial assay, kit | Quick adhesive cement system (Metabond) | Parkell | Cat. # S380 | |
| Chemical compound, drug | Phenylephrine | West-Ward Pharmaceutical | NDC 0641-6229-01 | |
| Other | Lipopolysaccharide (LPS) from *Escherichia coli* 0111:B4 | Sigma-Aldrich | Cat. # L-4130 | |
| Commercial assay, kit | TNF-α mouse ELISA kit | Invitrogen | Cat. # 88-7324-88 | |

*Continued on next page*

*Continued*

| Reagent type (species) or resource | Designation | Source or reference | Identifiers | Additional information |
|---|---|---|---|---|
| Antibody | Rabbit anti-neurofilament heavy polypeptide | Abcam | Cat. # ab8135 RRID:AB_306298 | 1:500 |
| Antibody | Goat anti-rabbit Alexa 488 secondary antibody | Invitrogen | Cat. # A-11008 RRID:AB_143165 | 1:500 |
| Commercial assay, kit | Trichrome stain kit | Abcam | Cat. # ab150686 | |

## Electrode preparation

150 µm MicroLeads cuff electrodes (MicroLeads Neuro, Gaithersburg, MD) and 100 µm CorTec microsling cuffs (CorTec, Freiburg, Germany) were commercially fabricated and used for cuff preparation (*Figure 1A*). MicroLeads cuffs are constructed with medical-grade assembly techniques using biocompatible materials, largely silicone, polyimide, and platinum iridium, and utilize a self-closing mechanism. CorTec microsling cuffs are also constructed with biocompatible materials including a silicone base, platinum iridium contacts, and a parylene C coating, and utilize a buckle-like closing

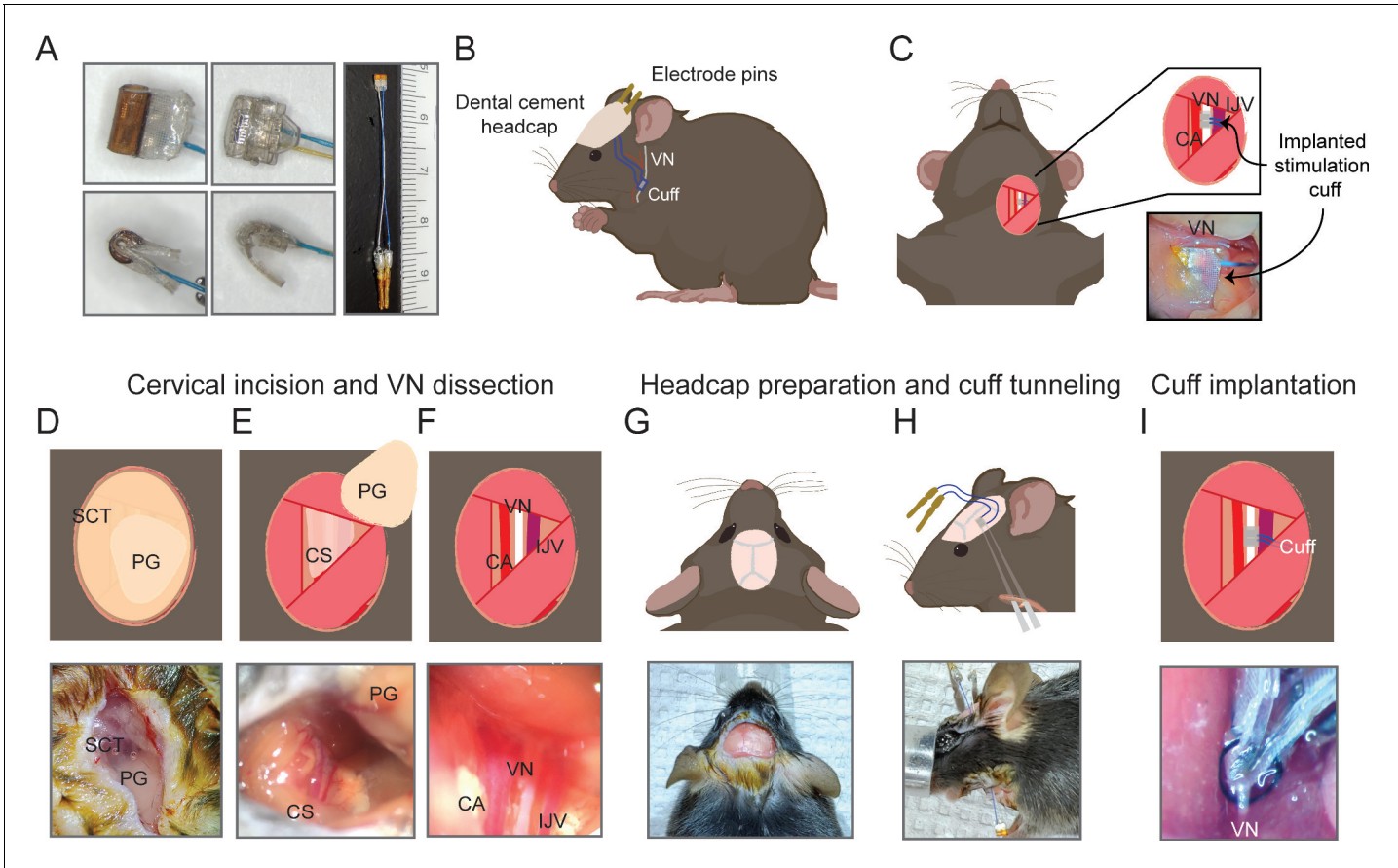

**Figure 1.** Surgical procedure for long-term implantation. (A) Lead wires of cuff electrodes produced by MicroLeads or CorTec are cut to a length of 2.5–3.0 cm and soldered to gold pins (right panel, MicroLeads). Front-facing (upper panels) and side (lower panels) view of 150 µm MicroLeads and 100 µm CorTec cuff electrodes. (B, C) Overview of implant, headcap with pins and location of the vagus nerve (VN) cuff. The cuff is implanted on the left cervical VN. (D) A 1 cm ventral incision is made about 0.5 cm lateral to the sternal notch, exposing subcutaneous tissue (SCT) and the parotid gland (PG). (E) SCT is bluntly dissected freeing the PG, which is then retracted from view exposing the carotid sheath (CS). (F) The VN is bluntly dissected away from the carotid artery (CA) and the internal jugular vein (IJV). (G) The scalp is incised to expose lambda and bregma. (H) A subcutaneous tunnel is created from skull base to cervical incision site, either between the eye and ear (depicted) or directly caudal to the ear. (I) The cuff is tunneled under the sternomastoid muscle and implanted on the VN. Pins are finally secured to the skull with dental cement. Created with BioRender.com.

mechanism. These bipolar platinum-iridium micro-cuff electrodes were soldered to gold sockets after cutting the lead wires to a length of 2.5–3.0 cm (*Figure 1A*). Electrical impedance was measured in saline for each electrode at 1 kHz using MicroProbes Impedance Tester (MicroProbes, Gaithersburg, MD). For sterilization, the soldered electrodes were submerged in 0.55% ortho-phthaladehyde solution (Cidex OPA, Advanced Sterilization Products, Irvine, CA) or 70% ethanol for 15 min, rinsed four times with sterile saline, and if Cidex used sonicated in saline for 5 min as a final rinse.

## Implantation procedure

Male C57BL/6 mice were purchased from Charles River Laboratories (Wilmington, MA) at the age of 8–12 weeks. Animals were housed under 12 hr light/dark cycle with ad libitum access to food and water. All animal experiments complied with relevant ethical guidelines and were approved by the Institutional Animal Care and Use Committee of the Feinstein Institutes for Medical Research (protocol numbers: 2016-029, 2017-010, and 2019-010) and University of Colorado Anschutz Medical Campus (protocol number: 00238). The surgery features an implantation of a cuff electrode on the cervical VN with lead wires that are tunneled to a headcap secured with dental cement to the skull (*Figure 1B, C*).

Mice were placed on a heated surgical platform equipped with a dissecting microscope in the supine position under isoflurane anesthesia at 4% induction and 1.5% maintenance. Hair over the neck area was removed using a depilatory cream (Nair, Church and Dwight, Ewing, NJ) or with an electric shaver and skin disinfected using alternating swabs of betadine and 70% ethanol. A 1 cm vertical incision was made starting at the level of the sternal notch about 0.5 cm left of midline (*Figure 1D*). The left lobe of the parotid (salivary) gland (PG) was bluntly dissected away from subcutaneous tissue (SCT) (*Figure 1D, E*). The muscles forming the anterior triangle were exposed providing a window to view the carotid sheath (CS) housing the carotid artery (CA), VN, and internal jugular vein (IJV). Next, the surgical field was expanded by retracting the sternomastoid muscle laterally using a magnetic fixator retraction system (Fine Science Tools, Foster City, CA) to ensure adequate access to the sheath. Using blunt dissection, the CS was isolated and the VN carefully dissected away from the fibrous connective tissue (*Figure 1F*). The incision site was temporarily closed, and the mouse was turned to the prone position to access the skull. The head was shaved and disinfected with alternating swabs of betadine and 70% ethanol. After application of a local analgesic, a fold of skin over the center of the skull was removed exposing bregma and lambda. The exposed skull was treated with three alternating rinses of hydrogen peroxide and saline, while gently scoring the skull surface with a sterile scalpel in between rinses (*Figure 1G*). After drying the skull with compressed air, acrylic dental cement (Metabond, Parkell, Edgewood, NY) was applied to the right half of the exposed area and allowed to set for 5 min. Using blunt dissection at the left edge of the scalp incision, a subcutaneous tunnel was created between the animal's eye and ear down to the ventral neck incision location (*Figure 1H*). Alternatively, the subcutaneous tunnel can be performed from caudal to the dorsal ear at the base of the skull, or through a 1 cm midline incision on the dorsal neck, to the ventral incision. The preformed subcutaneous tunnel was accessed using blunt dissection at the lower edge of the neck incision, and a pair of fine straight forceps were used to pull the electrode cuff subcutaneously into the neck area (*Figure 1H*). The cuff was further tunneled under the sternomastoid muscle and placed next to the VN. The VN was gently lifted with a micro surgical hook and the cuff passed underneath it with care taken to avoid excessive manipulation. The nerve was then placed into the cuff close to the nerve's original anatomic position and the cuff-specific closing mechanism engaged (*Figure 1I*). To confirm successful electrode placement, a brief stimulus was delivered through the externalized electrode leads to measure heart rate (HR) response. The neck incision was then closed with 6-0 nylon suture, and the animal turned to the prone position. The externalized connectors were held at an angle against the exposed part of the skull and acrylic dental cement (Metabond) applied covering the leads and part of the connectors and allowed to set for 5 min (*Figure 1B*). To seal the headcap to the skin, liquid surgical adhesive (Vetbond, 3M, Saint Paul, MN) was applied to the skin-cement junction. Alternatively, a six-channel pedestal (F12794, P1 Technologies, Roanoke, VA), surrounded by a 2 cm circle of polypropylene mesh (PPKM404.35, Surgical Mesh, Brookfield, CT), was placed into the dorsal midline neck incision after forming a subcutaneous pocket. The incision superior and inferior to the pedestal was closed with absorbable suture and surgical clips. The mice were moved to clean, warmed cages and

monitored until conscious and mobile. The surgical procedures were carried out under strict aseptic conditions, and animals were supplemented with warm saline intra- and postoperatively. Sham surgery animals underwent the same procedures, including nerve isolation and manipulation and subcutaneous tunneling, without the creation of a headcap. In some experiments, animals' body weights and food intake were recorded daily post-implantation for at least 2 weeks. To monitor food consumption, singly housed animals were provided with a measured amount of laboratory chow (about 13 g) in a Petri dish each day, which was collected and weighed the next day and replaced with a fresh amount.

For awake stimulation experiments, mice were instrumented with implanted ECG electrodes to measure heart rate threshold (HRT) in conscious animals. Following the surgical approach described above, three platinum wires were tunneled subcutaneously along the cuff leads from the skull to the ventral neck. The left ECG lead was tunneled subcutaneously through a 1 cm incision at the left costal margin and the exposed part fixed to the underlying muscle with 6-0 nylon suture. The right ECG lead was tunneled subcutaneously from the neck incision and sutured to the pectoralis muscle. The ground ECG lead was imbedded in the neck between the right lobe of the salivary gland and the skin. The ECG and cuff leads were connected to a multi-channel nano-connector (Omnetics Connector Corporation, Minneapolis, MN) and cemented to the skull as described before.

## Nerve stimulation and physiological monitoring

Validation experiments were carried out in several cohorts of mice by three research groups (Zanos group at Feinstein Institutes, Welle group at University of Colorado Anschutz Medical Campus, and Tracey group at Feinstein Institutes). Electrode functionality was evaluated by the ability to induce a decrease in HR during stimulation in anesthetized animals. HRT was defined as the minimum current intensity required to elicit an ~5–15% reduction in HR using a stimulus train of 300 bi-phasic, charge-balanced, square pulses at a pulsing frequency of 30 Hz with short (100 µs) or long (500–1000 µs) pulse widths (PWs). In most cases, HRT was initially determined with short PWs, which was changed to long PWs (500/600 µs and finally to 1000 µs whenever HRT exceeded 2 mA); in four mice, HRT was determined with both short and long PWs over several sessions. In one cohort, mice were tested on 3–7 days during the first week post-implantation, then once or twice weekly thereafter, whereas another two cohorts were tested less frequently or regularly. During testing sessions, anesthetized mice were instrumented with ECG electrodes and a nasal temperature sensor (IT-23 microprobe, Physitemp Instruments, Clifton, NJ) to measure ECG and nasal air flow and calculate HR and breathing rate (BR). The physiological signals were amplified using a biological amplifier (Bio-Amp Octal, ADInstruments, Colorado Springs, CO) for ECG and Temperature Pod (ADInstruments) for nasal temperature and digitized using PoweLab 16/35 (ADInstruments). The digital signals were then streamed to a PC running LabChart v8 (ADInstruments). VNS was delivered by a rack-mounted stimulus generator (STG4008, Multichannel Systems, Reutlingen, BW Germany). In a fourth cohort of animals, stimulation response was defined as a reduction in HR measured with an infrared paw sensor (Mouse Stat Jr, Kent Scientific) or respiratory rate (measured visually) in response to a stimulus train of 0.2–1 mA intensity, short PW, and 30 Hz frequency. Stimulation failure occurred when there was no response in either HR reduction or BR alterations. Other failure modalities included headcap failure. Cuff functionality was tested regularly within the first 14 days. Thereafter, a subset of the mice was selected for additional stimulation testing on a per-needed basis for further experiments.

In awake experiments, animals implanted with ECG leads were gently restrained and connected to a commutator (P1 Technologies, Roanoke, VA) that interfaced with the stimulus generator and the bio-amplifier; HRT was determined as described above. Intensity at maximum charge injection capacity (CIC) was calculated using the average reported value of CIC for platinum iridium (50–150 µC/cm$^2$) (*Cogan, 2008*; *Merrill et al., 2005*) applied to the implanted electrode surface area (0.00474 cm$^2$) for short and long PWs.

## Baroreflex assessment

Implanted and naive mice were anesthetized and placed on a warmed surgical platform in the supine position and instrumented with ECG leads. The right external jugular vein was isolated by blunt dissection and two sutures were placed rostrally and caudally. The rostral suture was ligated to prevent bleeding. After occluding blood flow by pulling on the caudal suture, a small incision was made in

the jugular vein and a 1 French catheter (Instech Labs, Plymouth Meeting, PA) was carefully advanced into the vessel after removing the caudal suture. A small amount of saline was injected to confirm the catheter was functional. The right CS was then exposed. A 1.4 French pressure catheter (SPR-671, Millar, Houston, TX) was carefully advanced into the artery using the same technique described for the jugular vein. Once the two catheters were in place and confirmed functional, 100 µl of phenylephrine (25 µg/kg) supplemented with heparin (7 U/ml) in saline was injected into the jugular vein over 7 s (*Fleming et al., 2013*) and pressure and HR monitored. Systolic and diastolic pressure and ECG signals were amplified using Bio-Amp Octal (ADInstruments) as described before. To calculate the baroreflex sensitivity index, a 10 s window around the peak systolic blood pressure (BP) was identified to calculate the systolic BP and corresponding HR after phenylephrine injection. Baseline values were calculated from a 10 s window immediately before the injection.

## LPS endotoxemia challenge

Lipopolysaccharide (LPS) from *Escherichia coli* 0111:B4 (Sigma-Aldrich, St. Louis, MO) was dissolved in saline and sonicated for 30 min before administration. LPS doses were determined empirically to produce physiological levels of TNF as described in *Caravaca et al., 2019*. In one set of experiments, performed by the Tracey group (Feinstein Institutes), 8-week-old mice (n = 12) were implanted with a left VN cuff. On day 9–17 post-surgery, VNS or sham stimulation was delivered twice (once in the morning and once in the evening) under light anesthesia using 1 mA intensity at 250 µs PW and 30 Hz frequency for 5 min. On the following day, mice were administered LPS (0.7 mg/kg, i.p.) 5 hr after receiving a third dose of VNS or sham stimulation. In another set of experiments, performed by the Zanos group (Feinstein Institutes), 8-week-old mice were implanted with a left VN cuff. HRT was determined at least 5 days before endotoxemia to avoid any long-lived VNS effects. 2–6 weeks post-implantation, animals were anesthetized and received either sham stimulation or VNS at HRT intensity using 250 µs PW and 10 Hz frequency for 5 min. LPS was administered to mice (0.1 mg/kg, i.p.) 3 hr after stimulation. In both sets of experiments, blood was collected by cardiac puncture 90 min post-LPS injection and left to clot for 1 hr at room temperature. The blood samples were then centrifuged at 2000 x*g* for 10 min and serum collected for TNF determination by ELISA (Invitrogen, Carlsbad, CA) following the manufacturer's instructions. Serum samples were assayed in duplicate for each animal.

## Histology and immunohistochemistry

Mice with long-term implants of at least 2 weeks old (n = 4) or naive controls were anesthetized, and the implant site carefully exposed to locate and isolate the nerve relative to anatomical landmarks. Mice were then euthanized, and the nerve fixed in place by filling the incision site with 10% buffered formalin for about 30 min. The nerve, along with the cuff in implanted animals, was then explanted and kept in 10% formalin overnight. The following day, the cuff electrodes were removed under a dissecting microscope, and the nerve samples grossed and prepared for paraffin embedding or frozen sectioning. Serial cross-sections of the tissue specimens were obtained at 5 µm thickness using a microtome and subsequently deparaffinized in preparation for staining. In some experiments, mice with chronic implants of at least 4 weeks (n = 5) were euthanized and segments of the neck were excised and fixed in 10% buffered formalin for at least 2 weeks. Fixed segments were then prepared for frozen sectioning. Serial cross-sections of the tissue specimens were obtained at 50 µm thickness using a cryostat. Standard immunohistochemical protocols were followed to stain the mounted sections for neurofilament (*Crosby et al., 2016*). Briefly, sections were rinsed with 1× Tris-buffered saline (TBS) then blocked for 1 hr using 1% normal goat serum and Triton X-100 (Sigma Aldrich) in TBS. Sections were then incubated with a primary antibody against neurofilament (1:500, ab8135, Abcam, Cambridge, MA) overnight at 4°C. The following day, sections were rinsed and incubated with goat anti-rabbit Alexa 488 secondary antibody (1:500, Thermo Fisher, Waltham, MA) for 2 hr at room temperature. Following incubation, stained slides were rinsed three times with TBS buffer then mounted with Fluoromount-G (Thermo Fisher). Images of the VN were obtained with ×100 magnification using a Keyence BZ-X810 fluorescence microscope (Keyence, Osaka, Japan). Hematoxylin and eosin (H&E) and Masson's trichrome (Trichrome Stain Kit, Abcam) stains were performed using standard protocols. In neck block sections, the left VN was identified either within the tissues covering the upper margin of the cuff or in the most anterior part of the neck adjacent to the cuff.

## Statistical analysis

Pearson correlation was used to characterize the relationship between implant age and HRT, and between implant age and electrical impedance; p-values less than 0.05 were deemed statistically significant. Student's t-test (or Mann–Whitney U for non-Gaussian variables) was used to compare between two means with Bonferroni correction for multiple comparisons; p-values less than 0.05 were deemed statistically significant.

## Results

### A surgical procedure to interface with the VN in mice

We first set out to design a surgical process that allows for successful long-term implantation of a micro-cuff electrode onto the mouse cervical VN. Due to the small size of the nerve (~100 µm in diameter) and cuff, this required a carefully considered protocol. We first optimized our surgical approach to isolate the CS with minimal tissue injury by employing blunt dissection using a set of fine forceps and a surgical hook (*Figure 1E*). Retracting the sternocleidomastoid muscle and parotid gland is critical in obtaining an adequate view of the sheath before VN isolation. Following the same principle, the VN was carefully dissected along its length using fine blunt forceps after identifying the pulsating internal CA just posterolateral to the trachea (*Figure 1F*). Electrode tunneling is another critical step that requires minimizing lead wire travel distance and mechanical strain, while maintaining stability. We found that subcutaneous tunneling around the neck or directly between the eye and ear (*Figure 1H*) both result in equally successful implants. Further, tunneling deep to the sternocleidomastoid muscle helps align the cuff on the same plane as the nerve and minimize lateral tracking of the cuff by providing a muscular border. We also found that maintaining a front-facing cuff orientation (*Figure 1A*) as the cuff is tunneled to the vagus nerve ensures easy placement and prevents the cuff from pulling at or twisting the nerve. Careful adjustment of the cuff orientation intraoperatively to achieve minimal anatomic disruption improves surgical outcomes. Potential mechanical damage is further reduced by using cuffs slightly larger (100–150 µm) than the diameter of the vagus to prevent compression and by incorporating coiled wires to reduce mechanical strain. Moreover, construction of a robust headcap contributes to the stability and longevity of the implant and results in minimal headcap failures (n = 1) (Figure 3F). Careful preparation of the skull, including complete tissue removal, and adequate scoring and drying helps bind the cement to the skull surface and prevent infections. Also, the silicone construction of the cuff shell as well as including adequate distance between the edges of the cuff and stimulating electrodes reduces current leakage to surrounding tissues.

### VNS through the long-term implant elicits changes in HR and BR

The cervical VN comprises parasympathetic motor and visceral sensory fibers that regulate many physiological functions including HR and breathing (*Berthoud and Neuhuber, 2000*; *Agostoni et al., 1957*; *Chang et al., 2015*). To characterize the physiological outcomes of stimulation through the implanted micro-cuff, we stimulated the VN with increasing current intensity while measuring stimulus-elicited changes in HR and BR in animals under isoflurane anesthesia. VNS produces decreases in HR as well as changes in BR (*Figure 2A*). The magnitude of HR reduction is dependent on current intensity (r = 0.8971, p=0.0062) (*Figure 2B*), whereas BR shows more variable responses, including slowing down and acceleration of breathing (*Figure 2A*, *Figure 3A*). In conscious mice (n = 2), VNS produces comparable dose-dependent HR responses (*Video 1*, *Figure 2— figure supplement 1*). Animals receiving awake VNS do not show any signs of distress or visible changes in BR.

### Longitudinal changes in implant functionality

To assess the longitudinal functionality of each implant, we determined HRT over time, defined as the minimum current intensity of a stimulus train (300 pulses at 30 Hz) required to elicit an approximately 5–15% decrease in HR. VNS delivered using the implant elicits drops in HR and changes in breathing for up to 8 weeks post-implantation (*Figure 3A*). Initial HRT values determined with short PW were variable among animals (range = 30–400 µA, mean = 156, SD = 118, n = 20 mice) and increased over the first week post-implantation (Pearson r = 0.89, p = 0.0064, n = 20 mice)

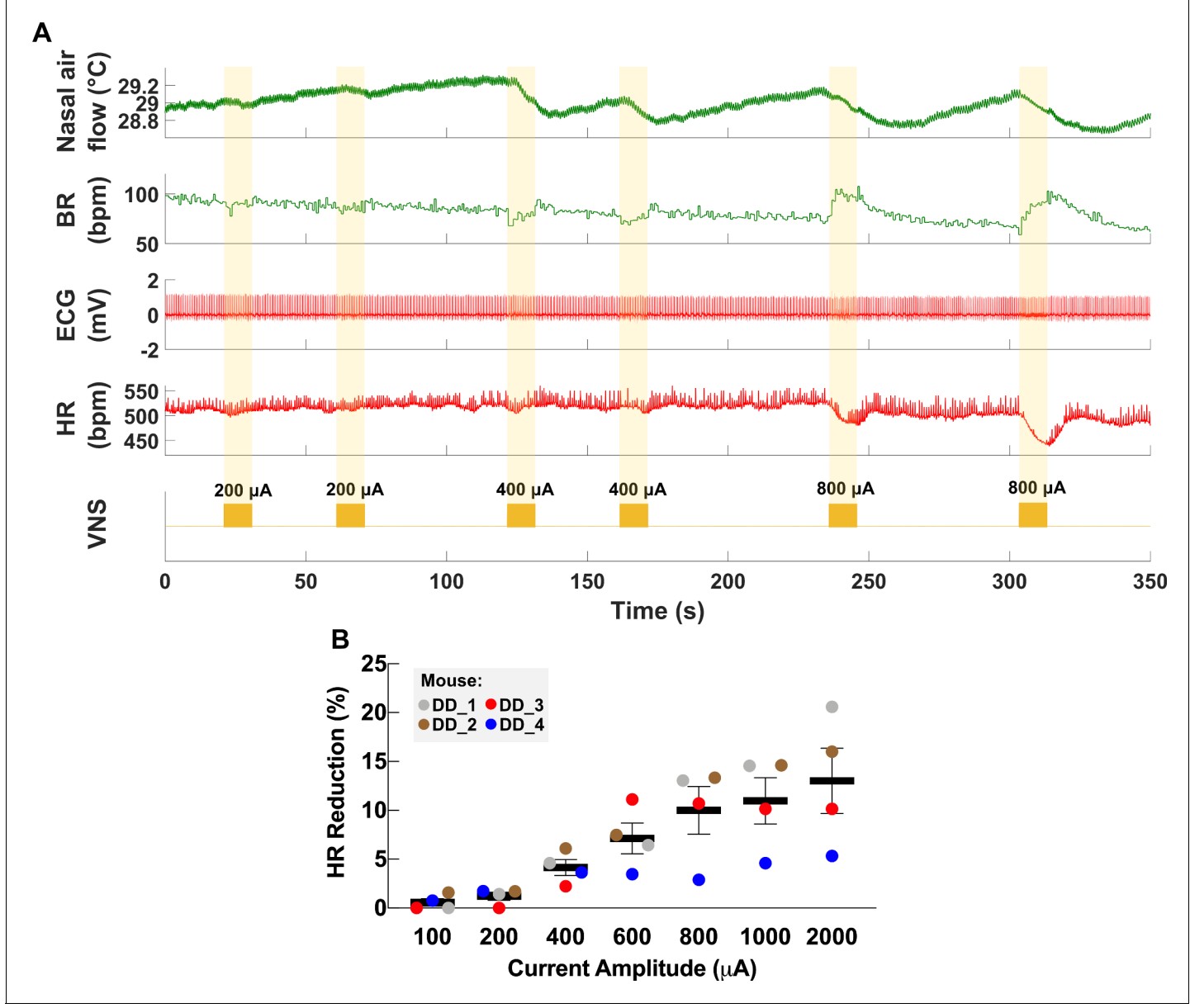

**Figure 2.** Dose-dependent physiological responses to vagus nerve stimulation (VNS). (**A**) Representative traces from a chronically implanted mouse showing nasal air flow (top panel) and extracted breathing rate (BR, second panel), and ECG (third panel) and extracted heart rate (HR, fourth panel). Trains of VNS of increasing intensity from 200 to 800 µA (fifth panel, yellow traces) cause BR and HR responses with increasing magnitudes. (**B**) Percentage of HR reduction as a function of VNS intensity in four chronically implanted mice (parameters: short pulse width, frequency 30 Hz, duration 5 s).

The online version of this article includes the following source data and figure supplement(s) for figure 2:

**Source data 1.** Source data file (.xlsx) containing heart rate (HR) measurements at baseline and during vagus nerve stimulation (VNS) at different intensities used to create *Figure 2B*.

**Figure supplement 1.** Vagus nerve stimulation (VNS) in a conscious mouse.

(*Figure 3—figure supplement 1A*), whereas HRT values determined with long PWs did not change significantly with time (r = −0.22, p NS, n = 26 mice) (*Figure 3B*). HRT values with short PW were 54% greater on average than those with long PW, and that relationship was maintained over time (*Figure 3C*). Pre-implantation impedance values did not correlate with initial HRT (r = 0.34, p NS, n = 15 mice) (*Figure 3—figure supplement 1B*), and bipolar electrical impedance did not increase over the period of testing (r = −0.68, p = 0.0415, n = 29) (*Figure 3D*). Interestingly, there was no

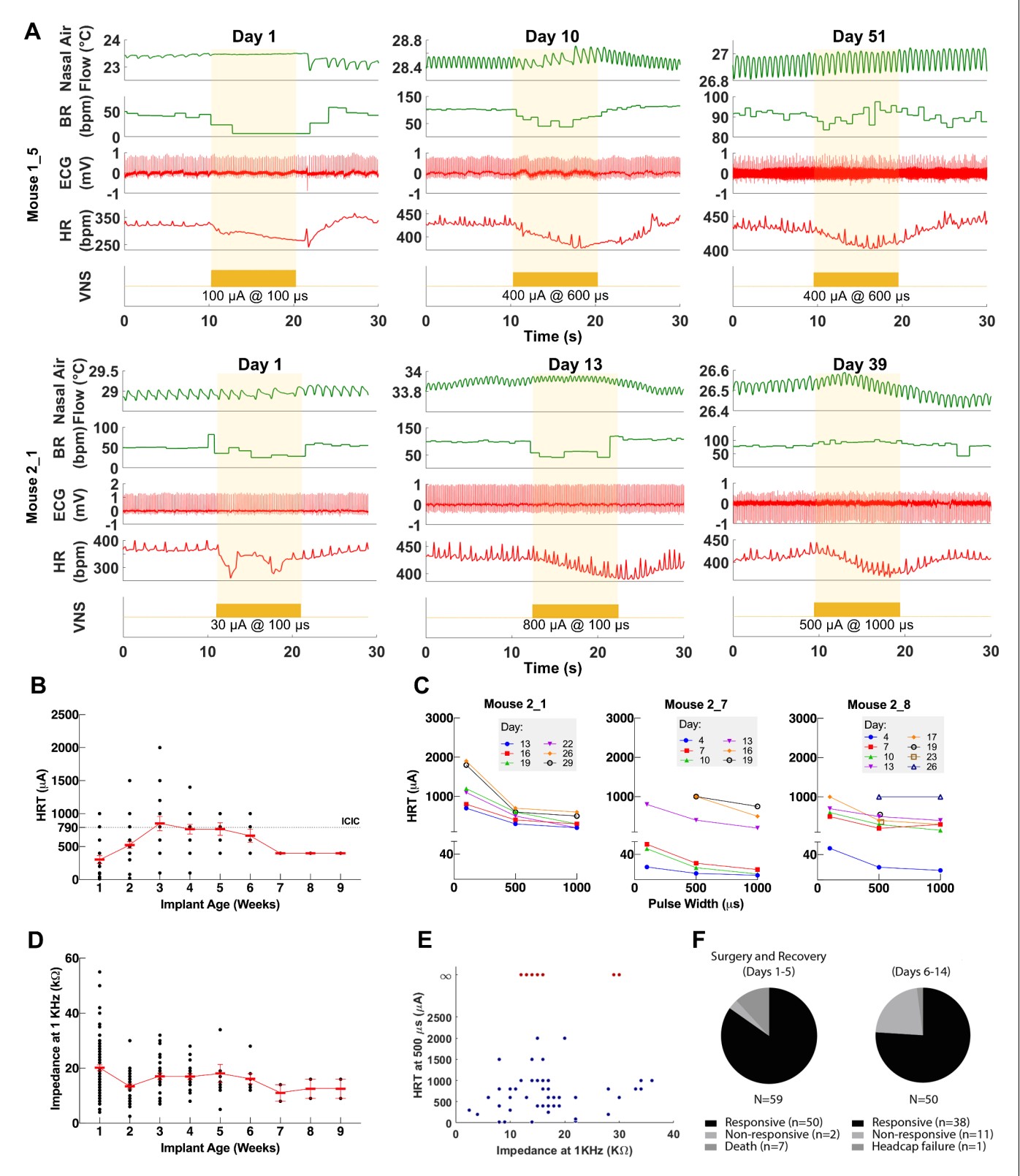

**Figure 3.** Longitudinal changes in heart rate threshold (HRT) and electrode impedance of long-term implants. (A) Examples of physiological responses to vagus nerve stimulation (VNS) in two mice, showing changes in breathing rate (BR; green trace) and heart rate (HR; red trace) elicited by a train of VNS with HRT intensity, as determined on that day of testing (yellow trace, with stimulus parameters shown). HRT is defined as the stimulation intensity required to produce an ~5–15% decrease in HR. (B) HRT values vs. implant age (n = 26 mice), where multiple measurements were grouped together

*Figure 3 continued*

under the corresponding week. Horizontal gray dotted line indicates intensity corresponding to maximum charge injection capacity (ICIC) with 600 μs pulse width (PW), as calculated for these electrodes. (C) HRT values determined with VNS trains of 0.1, 0.5, and 1 ms-wide pulses at different implant ages in three mice. (D) Electrical impedance measured at different implant ages (n = 29 mice). (E) HRT values plotted against electrode impedance from individual measurements performed during a 40-day period post-implant. 'Infinity' HRT values (data points in red) indicate implants that did not produce a HR response up to 2 mA at 2 ms PW. Pearson correlation was 0.05 (p NS). (F) Surgical success rates for one of the tested cohorts of animals. The online version of this article includes the following source data and figure supplement(s) for figure 3:

**Source data 1.** Source data file (.xlsx) containing pre-implantation and longitudinal impedance and heart rate threshold (HRT) values from cohorts 1–3.
**Figure supplement 1.** Dependence of heart rate threshold (HRT) on electrode impedance and implant age.

correlation between changes in electrical impedance and HRT values (r = 0.05, p NS); in some mice, nonfunctional cuffs continued to have relatively low impedance values despite their inability to induce a physiological response (*Figure 3E*). Implant failure occurred more frequently in earlier compared to later cohorts: percentage of mice with functional implants at 4 weeks post-implantation increased from 40% in cohort 1 to 90% in cohort 3 (*Table 1*). In a separate group of animals, implants were 96% functional during the first 5 days after surgery (50/52 mice, excluding deaths during surgery); functional implants were tested again in days 6–14, with a success rate of 76% (38/50) (*Table 1* and *Figure 3F*). In a random subset of those implants, 17/18 and 11/13 were functional in the 15–30 and 30+ days period, respectively (*Table 1*). Overall, electrode failures occurred during the first 2 weeks post-implantation and implant functionality stabilized thereafter (*Table 1*).

## Long-term implantation does not impact vagally mediated reflexes

The VN modulates several vital bodily functions via reflexes, including appetite, BP, and respiration (*Paintal, 1973*). To demonstrate that long-term cuffing of the VN does not affect these reflexes, we evaluated the implanted animals' weight change and food intake, baroreflex, and breathing reflexes. The change in body weight of implanted animals during the first and second weeks post-implantation is not different than sham surgery controls (*Figure 4A*, *Figure 4—figure supplement 1A*). Further, the average food intake during the first 2 weeks is similar in both groups (*Figure 4B*, *Figure 4—figure supplement 1B*) and within the range of reported daily average intake in healthy animals (*Bachmanov et al., 2002*). Implanted mice do not exhibit elevated levels of serum TNF 2–3 weeks post-implantation (*Figure 4—figure supplement 1C*). To evaluate the vagal component of the baroreflex, we injected implanted and naive animals with phenylephrine, a vasopressor that increases BP, and recorded reflexive changes in HR (*Figure 4C*). Both implanted and naive animals have similar HR at baseline, with a similarly significant decrease upon phenylephrine injection (*Figure 4C, D*). Further, the baroreflex sensitivity index, expressed as the ratio of heart rate change (ΔHR) to systolic blood pressure change (ΔSBP), is not significantly different between the two groups (*Figure 4E*). We also evaluated whether long-term cuffing affected vagally mediated breathing reflexes (e.g., Herring–Breuer reflex; *Chang et al., 2015*) by evoking breathing changes with electrical stimulation. We found that mice with long-term implants exhibit changes in breathing (*Figure 4F*, *Figure 4—figure supplement 1D*) similar to those induced in acute VNS experiments (*Figure 4—figure supplement 1E*).

## VNS using the long-term implant inhibits TNF release in endotoxemia

Acute VNS decreases serum TNF levels in acute inflammation by modulating the immune response via a neuroimmune mechanism termed

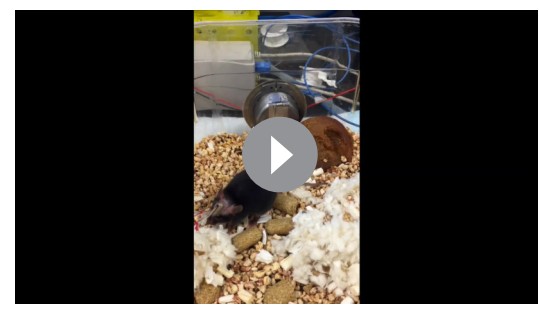

**Video 1.** Vagus nerve stimulation (VNS) in a conscious mouse. Video clip showing a conscious mouse with a long-term VN implant and ECG leads, connected to a commutator and receiving VNS on post-implant day 15. The screen shows heart rate (HR; green trace) and a stimulation event (purple trace). VNS occurs at the 23 s time point.
https://elifesciences.org/articles/61270#video1

**Table 1.** Functional implants across time in several animal cohorts.

Implants were tested in four cohorts. In cohorts 1–3, implant functionality was determined based on heart rate threshold, and implant failure was defined as the absence of a physiological response upon stimulation with 3 mA or higher on three consecutive testing sessions. In cohort 4, functionality was determined based on a reduction in heart rate or breathing rate and failure was defined as absence of response with 1 mA on one occasion.

| | Functional cuffs | | | |
| --- | --- | --- | --- | --- |
| Days post-implantation: | 1–5 | 6–14 | 15–29 | 30+ |
| Cohort 1 (n = 10) | 8/10 (80%) | 6/10 (60%) | 4/10 (40%) | 4/10 (40%) |
| Cohort 2 (n = 10) | 10/10 (100%) | 7/10 (70%) | 7/10 (70%) | 6/10 (60%) |
| Cohort 3 (n = 9) | 9/9 (100%) | 8/9 (90%) | 8/9 (90%) | – |
| Cohort 4 (n = 52) | 50/52 (96%) | 38/50 (76%) | 17/18* | 11/13* |

*Group is a randomly selected subset of the (6–14 days) functional implants (n = 38).

the IR (*Borovikova et al., 2000*). To test whether our long-term implant can produce a similar effect, we used it to deliver VNS in an LPS endotoxemia model of acute inflammation using a set of parameters that have been shown to inhibit TNF release in acute experiments in mice (*Caravaca et al., 2019*). Mice with 9- to 17-day-old implants received three VNS doses (1 mA intensity, 250 µs PW, 30 Hz frequency, 5 min duration): two doses administered 1 day prior to LPS administration and one dose on the following day, 5 hr before LPS or vehicle (saline) administration (*Figure 5A*). We found that VNS significantly decreases serum TNF in stimulated animals compared to sham stimulation (*Figure 5B*). Notably, these parameters did not usually induce a change in HR during stimulation (*Figure 5C*). In another experiment, we tested whether one-time VNS could decrease serum TNF. Mice with either 6-week-old or 16- to 19-day-old implants received VNS (HRT intensity, 250 µs PW, 10 Hz frequency, and 5 min duration) or sham stimulation, 3 hr before LPS administration (*Figure 5D*). Overall, we found that one-time VNS does not produce a significant decrease in TNF levels. However, out of the 14 stimulated mice, VNS produced a decrease in HR in seven animals, of which four exhibited more than ~40% decrease in serum TNF compared to sham-stimulated controls and animals with no HR response (*Figure 5E*). Mice that lacked a physiological response had TNF levels comparable to sham-stimulated controls.

## Long-term implantation does not induce significant nerve damage

Long-term efficacy of peripheral nerve implants could deteriorate due to direct nerve damage or reaction of surrounding tissue to the electrode (*Anderson et al., 2008*; *Tyler and Durand, 2003*). To determine the impact of these processes in our long-term implants, we collected cuffed and non-cuffed left VNs from implanted mice at ~2–6 weeks post-implantation for gross and histological analysis; naive mice were used as controls. The implant site appeared healed and exhibited moderate tissue growth encompassing the lead wires and cuff surfaces in animals with both functional and non-functional implants (example from 12 days post-implantation shown in *Figure 6A*). Histological analysis of explanted cuffed left nerves revealed preserved nerve fibers compared with non-cuffed left nerves from naive controls with no obvious axonal pathology or inflammation (*Figure 6B*, *Figure 6— figure supplement 1A, B*). The explanted nerves were surrounded by increased amounts of fibrotic tissue (*Figure 6C*) or exhibited thickened perineurium (*Figure 6B*, *Figure 6—figure supplement 1A*). In another group of animals (n = 6) in which we examined cross-sections of whole neck blocks just above the cuff margin 6 weeks post-implantation, histological analysis revealed similar preserved nerves (*Figure 6—figure supplement 1C*).

## Discussion

VNS is an emerging bioelectronic therapy with possible applications in many chronic diseases. However, its translational potential is hindered by the lack of a reliable long-term VNS implant in mice—the preferred species in the preclinical study of human diseases (*Vandenbergh, 2008*). Development of a simple, well-characterized, and reliable long-term VNS interface in mice will allow for

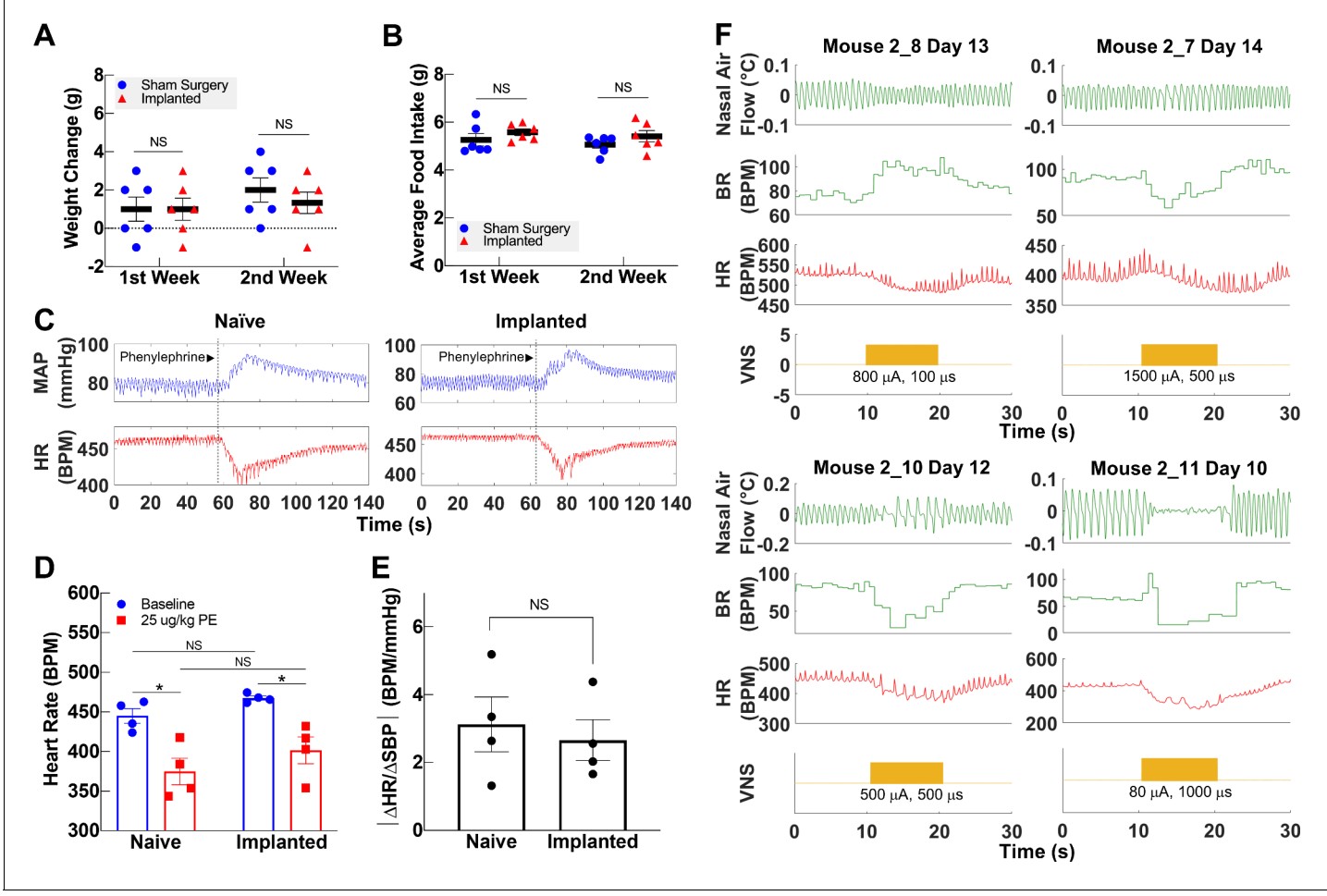

**Figure 4.** Vagally mediated reflexes in animals with long-term implants. (A) Weight change at first and second week post-surgery in mice subjected to sham surgery (n = 6) and in implanted mice (n = 6). (B) Average food intake during the first and second week post-surgery in sham surgery and in implanted mice (n = 6 in each group). (C) Example of baroreflex-mediated changes in heart rate (HR) in response to phenylephrine (PE)-elicited increase in blood pressure in a naive animal (left) and in an animal with a long-term implant (right). Traces showing mean arterial blood pressure (MAP; green trace) and HR (red trace); vertical line indicates time of PE injection. (D) HR in naive and implanted animals before and after PE injection. (E) Baroreflex sensitivity index in naive and implanted animals (n = 4 in each group). Index is calculated as the absolute value of the change in HR (ΔHR) over the change in systolic blood pressure (ΔSBP), before and after PE injection. (F) Examples of HR (red trace) and BR (green trace) changes in four mice with long-term implants showing responses to vagus nerve stimulation (VNS); BR responses include rapid, shallow breathing (upper left), and slowing down (upper right, lower left), or cessation of breathing (lower right) during VNS. Data is presented as mean ± SEM; NS = not significant, *$p<0.05$ by Student's t-test with Bonferroni correction for multiple comparisons.

The online version of this article includes the following source data and figure supplement(s) for figure 4:

**Source data 1.** Source data (.xlsx) file containing heart rate and blood pressure changes after phenylephrine injection used to calculate baroreflex sensitivity index.

**Figure supplement 1.** Examples of intact vagal reflexes in animals with chronic implants.

standardized assessment of long-term VNS efficacy, as well as potential adverse effects, in various models of chronic disease. It will also allow mechanistic studies of autonomic tone alterations that might accompany long-term VNS. Here, we describe a surgical approach to permanently implant a micro-cuff electrode onto the mouse cervical VN. In addition, we provide a systematic method to assess its functionality and standardize stimulation dosing over time. Our data demonstrate a robust and reproducible interface with the VN capable of producing characteristic physiological responses for at least 4 weeks post-implantation, while causing no large-scale axonal damage. We also provide functional evidence of intact vagal reflexes and anti-inflammatory action.

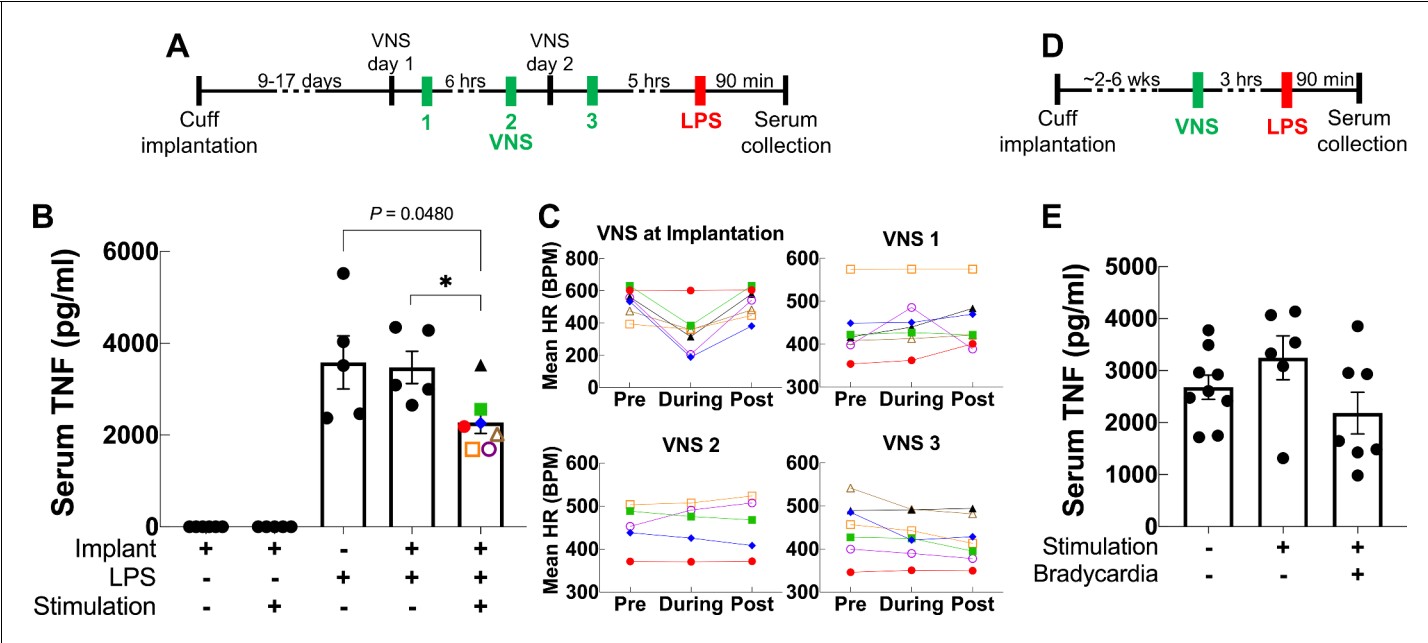

**Figure 5.** Vagus nerve stimulation (VNS) using the long-term implant inhibits TNF release in endotoxemia. (**A**) Mice with 9- to 17-day-old implants received three doses of VNS over 2 days using previously reported parameters (intensity 1 mA, pulse width [PW] 250 μs, frequency 30 Hz). Lipopolysaccharide (LPS) was administered 5 hr after the last VNS dose, and blood was collected 90 min post-LPS injection. (**B**) Serum TNF levels from implanted mice that received no LPS (saline) with sham or VNS (first and second bars), sham surgery mice that received LPS (third bar), and implanted mice that received LPS with sham or VNS (fourth and fifth bars). Data shown as mean ± SEM. *p = 0.0177, by Mann–Whitney with Bonferroni correction for multiple comparisons. (**C**) Mean heart rate (HR) before, during, and after each stimulation event for endotoxemic mice that received VNS. Each line corresponds to a subject with matching shape and color in fifth bar in (**B**). (**D**) In a separate experiment, mice with 2- to 6-week-old implants (n = 22) received a single dose of VNS, or sham VNS, for 5 min (intensity at heart rate threshold [HRT], PW 250 μs, frequency 10 Hz). LPS was administered 3 hr after VNS and blood was collected 90 min post-LPS injection. (**E**) Serum TNF levels from mice that received sham stimulation (left bar), mice that received VNS without a HR response (middle bar), and mice that received VNS that elicited a HR response (right bar). Data shown as mean ± SEM. p NS (VNS with bradycardia vs. sham, and VNS with vs. without bradycardia) by Mann–Whitney with Bonferroni correction for multiple comparisons.

To maximize the applicability of our tool in various research programs carried out by teams with different areas of expertise, we tried to accomplish two goals: ease of assembly and use, and reproducibility. Assembly of the implant makes use of only off-the-shelf supplies and materials, including commercially available micro-cuffs and common physiological sensors. For example, determining HRT requires the use of a simple rodent heart monitor. The surgical technique, which reflects the aggregate experience of three research groups, was refined and simplified over the course of several animal cohorts. Finally, validation of the longitudinal performance of this implant by three research groups supports the reproducibility of this method when exercised by different investigators.

Previous research using VNS in mouse models of disease has been limited to acute, single-event, stimulation (*Huffman et al., 2019*; *The et al., 2007*; *Ji et al., 2014*). These studies, although of significant translational value, provide less insight into the possible role of VNS in the treatment of chronic conditions. Moreover, acute stimulation studies are carried out under anesthesia, which confounds the results due to the effects of some anesthetics, such as isoflurane, on decreasing vagal tone (*Marano et al., 1996*; *Picker et al., 2001*) and suppressing the immune response (*Cruz et al., 2017*). For these reasons, a long-term interface to deliver stimulation to the VN is needed. However, to date, reported attempts at a stable and functional interface with the VN in mice either focused on long-term neural recordings with no longitudinal stimulation data (*Caravaca et al., 2017*; *Falcone et al., 2020*), or provided insufficient evidence of long-term implant functionality and effectiveness (*Ten Hove et al., 2020*). For example, while *Caravaca et al., 2017* reported longitudinal neural recordings, VNS was delivered to a separate subset of animals 3 hr post-implantation in a terminal model of endotoxemia, essentially an acute procedure. *Falcone et al., 2020* used their long-term microwire interface to collect neural signals but did not provide longitudinal stimulus-elicited

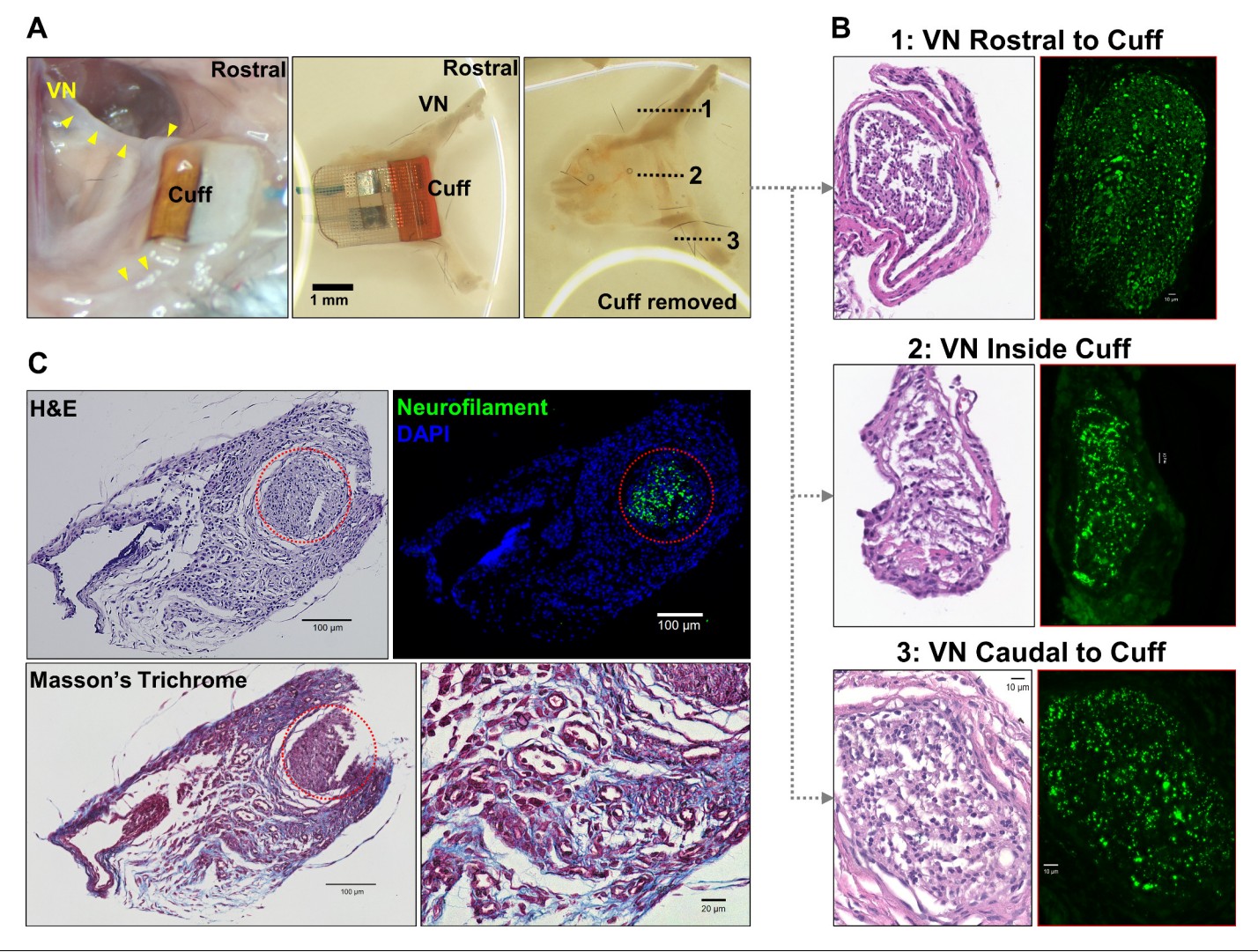

**Figure 6.** Gross anatomy and histology of cuffed nerve and surrounding tissues. (**A**) 3-week-old micro-cuff electrode upon explantation (right panel). The cuff was carefully removed with intact rostral and caudal portions (middle panel) and nerve sectioned at three levels (dotted lines, left panel). (**B**) Cross-sections of cuffed vagus nerve at corresponding levels shown in (**A**) stained with hematoxylin and eosin (H&E) (left panels) and for axons (neurofilament, green; right panels). (**C**) Explanted cuffed vagus nerve (red dotted circle, upper left panel, H&E) stained for axons (upper right, neurofilament, green; DAPI nuclear stain, blue), and collagen (lower panel, Masson's Trichrome).

The online version of this article includes the following figure supplement(s) for figure 6:

**Figure supplement 1.** Histology examples of cuffed and non-cuffed VNs.

physiological responses to demonstrate continued stimulation functionality. In a recent study by *Ten Hove et al., 2020*, VNS was delivered in mice for 12 days via a commercial cuff electrode to study its effects on ACE2 expression in intestinal cells. Although the authors provided histological evidence for an intact nerve structure, but without showing viable nerve fibers, they relied on visual observation of stimulus-elicited breathing and movement changes to confirm electrode functionality, an unreliable method for assessing nerve fiber engagement, as pain or irritation could produce a similar behavior. They also used electrical impedance to determine implant functionality, which, as shown in this report, is not well correlated to functional performance. Therefore, the reported lack of effect may be related to nonfunctioning implants that went undetected, given their method of testing. In this study, we established the feasibility of a long-term vagus interface in mice capable of producing stimulus-elicited responses for at least 4 weeks, a period that allows for therapeutic assessment in many models of chronic disease. Importantly, almost all animals achieved a full

recovery from the implantation procedure during the first week and maintained a stable implant after the second week. This is evident from the slow attrition of functional cuffs over time seen in randomly selected subsets of animals (*Figure 3F*). We also showed that this approach can be used to deliver dose-controlled VNS in conscious animals without repeated exposure to anesthetics, while ensuring consistent stimulation longitudinally and across different animals (*Video 1*). VNS in two conscious animals produced comparable changes in HR as those seen in anesthetized animals and implant longevity in those two animals ranged between 2 and 3 weeks. To our knowledge, this report is the first demonstration of chronic VNS in the mouse that provides longitudinal evidence of stimulus-elicited physiological responses.

VNS causes reduction in HR, mainly through activation of efferent cardioinhibitory fibers (*Buschman et al., 2006*). In a functional neural interface, increasing stimulation intensity leads to recruitment of more fibers, and hence a larger stimulus-evoked response (*Chang et al., 2020*; *Zaaimi et al., 2008*). This is consistent with our observation wherein changes in stimulation intensity resulted in a dose-dependent decrease in HR (*Figure 2B*). Additionally, higher stimulation intensities lead to the ordered recruitment of different fiber types according to size: from large A, to intermediate-size B, to small C fibers (*Blair and Erlanger, 1933*). The variable changes in BR we observed at different stimulation intensities (*Figure 2A, Figure 3A*) can be explained by activation of either A or C fibers, which differentially affect breathing in mice (*Chang et al., 2015*) and in rats (*Chang et al., 2020*). In addition, a functional nerve interface exhibits a characteristic relationship between intensity and PW (strength-duration): to produce a response, lower stimulation intensities are required at longer PWs, and vice versa (*Geddes and Bourland, 1985*). Our long-term implants produced HR responses at lower intensities with longer PWs, a relationship that remains consistent with time (*Figure 3C*). These findings indicate a robust electrode-tissue interface with the VN.

An effective approach to delivering long-term VNS must include standardized methods for verifying implant functionality and controlling stimulation dose over the course of treatment. To evaluate electrode performance across time, we used HRT as a quick and accessible measure of fiber recruitment in real time (*Yoo et al., 2016*). Initial threshold values were variable among animals and increased over the first week post-implantation in almost all mice (*Figure 3—figure supplement 1A*). The variability in baseline thresholds at the time of implantation is not explained by differences in pre-implantation impedance and is likely due to variability in electrode placement, which affects fiber engagement. The gradual increase in HRT over time can be attributed to fibrotic encapsulation of electrode surfaces, a process that evolves over days to weeks and is known to reduce the efficacy of implanted neural interfaces by increasing tissue resistivity and the distance between the nerve and the electrode contact surface (*Anderson et al., 2008*; *Grill and Mortimer, 1994*; *Farah et al., 2019*; *Vasudevan et al., 2017*). We observed increased tissue growth within the cuff, which, upon histological examination, comprised fibrotic tissue around the nerve (*Figure 6*). This may explain why longer PWs were more effective in eliciting HRT in aged implants: the effect of distance on activation threshold is weaker for long pulses than for short pulses (*Grill and Mortimer, 1995*). In addition to assessing performance, HRT was used as a method to estimate individualized stimulation doses. Since animals exhibit variable threshold values at baseline and across time, employing fixed parameters would result in variable fiber recruitment, inconsistent therapeutic dosing, and, possibly, undesirable off-target effects. This is of particular importance when VNS is delivered therapeutically, wherein standardized doses are desirable within and in between animals, and across time. Previous reports from rat and large animal models implemented a similar approach to adjust stimulation intensity using respiratory twitching (*Yoshida et al., 2018*; *Nishizaki et al., 2016*), and HR (*Huffman et al., 2019*; *Chapleau et al., 2016*; *Yu et al., 2014*) to standardize stimulation protocols, but not on a dose-by-dose basis. Notably, electrical impedance, which is commonly used as a measure of electrode integrity and performance (*Straka et al., 2018*), did not always correlate with changes in HRT on an individual implant basis as implants aged (*Figure 3E*). In fact, impedance tended to decrease over the period of implantation as HRT increased (*Figure 3B, D*). Moreover, several nonfunctional cuffs had relatively low impedance values (*Figure 3E*). For these reasons, we rely on HRT values as a reliable indicator of electrode integrity and a method to estimate stimulation dosage (*Qing et al., 2018*).

Implant longevity can be influenced by abiotic factors (*Prasad et al., 2014*; *Vasudevan et al., 2017*), such as lead breakage and electrode degradation. We observed three cases of lead wire breakage out of nine nonfunctional implants in three groups of animals. In all three cases, breakage

occurred at the junction between the lead wire and electrode, exposing a mechanically weak point that should be reinforced during manufacturing. Neural electrodes can degrade with long-term use (*Negi et al., 2010*). This process is accelerated at stimulation intensities that exceed the electrode's ability to transfer charge without undergoing irreversible damage (CIC) (*Negi et al., 2010*; *Merrill et al., 2005*). The damage imparted by exceeding this threshold is not limited to the electrode but can affect the nerve as well (*Cogan et al., 2016*). Although we did not examine explanted electrodes for morphological or electrochemical changes, we did take note of platinum-iridium's maximum charge capacity (*Cogan, 2008*), which was not exceeded in most animals (*Figure 3B, Figure 3—figure supplement 1A*). On the few occasions when ICIC was exceeded, it likely did not cause significant damage to the electrode as the stimulation events were brief (10 s) and limited in number. This becomes important when VNS is administered therapeutically in chronic models, wherein stimulation protocols may be employed on a daily basis for up to several weeks. In such settings, care should be taken not to exceed ICIC. Standardizing stimulation dose could prevent exceeding CIC while still delivering therapeutic stimulation. Mechanical forces generated by chronic cuffing can also damage nerves (*Agnew and McCreery, 1990*; *Larsen et al., 1998*; *Somann et al., 2018*). Our histological analysis revealed viable nerve tissue with no obvious fiber loss or axonal fragmentation. Therefore, damage to the nerve is unlikely to be a contributing factor to increased HRT over time. However, we interpret this preliminary analysis with caution, in part, due to the lack of small-fiber detail in these relatively thick sections (5 and 50 μm). Future studies will rely on higher resolution images with resin-embedding and electron microscopic analysis, which is the gold standard of peripheral nerve examination. Apart from electrode- and tissue-related factors, we found that surgical proficiency contributed greatly to implant success, which is evident from the increase in successful implantations over successive groups of animals (*Table 1*, cohorts 1–3, from earlier to later) completed over several months by an initially inexperienced surgeon. Over time, fine adjustments to electrode placement, surgical approach, and post-surgical care lead to higher success rates.

The VN maintains homeostasis by controlling many bodily functions, including appetite, respiration, and BP (*Paintal, 1973*). Chronic cuffing of the vagus could, in principle, cause damage to the nerve, thereby affecting these functions. For example, severing the VN causes weight loss in mice (*Kral and Görtz, 1981*) by affecting appetite and gastric transit times among other factors (*Dezfuli et al., 2018*; *Khound et al., 2017*). We did not observe weight loss or changes in food intake in our implanted animals compared with controls, which suggests that those vagal pathways were not affected (*Figure 4A, B*). The VN also mediates the afferent and the efferent-parasympathetic component of the baroreflex, a homeostatic mechanism that maintains BP within normal range by modulating HR (*Glick and Braunwald, 1965*). In an intact system, an increase in BP causes a reflexive decrease in HR by increasing vagal activity, which is consistent with our observations in implanted animals. The responsiveness of this system can be quantified using the baroreflex sensitivity index or the ratio of change in HR to change in BP, which we found in implanted mice to be comparable to naive controls (*Figure 4C, D*). The vagus also carries vital sensory information from the lung to regulate breathing (*Carr and Undem, 2003*). Recent evidence has shown that this is mediated by large, myelinated A fibers that cause slowing down of breathing and small, unmyelinated C fibers that cause rapid shallow breathing (*Chang et al., 2015*). These sensory pathways control respiration by reflexive activity that changes BR, such as the Herring–Breuer inflation and deflation reflexes (*Schelegle and Green, 2001*; *Siniaia et al., 2000*; *Yu, 2016*). In our studies, we were able to evoke similar changes in BR using VNS in anesthetized animals (*Figure 4F*). We, however, did not observe visible changes to BR during awake VNS. Even though there may still exist small alterations in BR that went unnoticed visually, changes in BR in awake animals may be more difficult to visually inspect due to movement, higher baseline BR compared to anesthetized mice, and compensatory mechanisms that are suppressed during anesthesia. Patients receiving VNS for epilepsy do not show changes in tidal volume or BR during wakefulness (*Banzett et al., 1999*), yet exhibit decreases in air flow when VNS is activated during sleep (*Murray et al., 2001*). Taken together, these data strongly suggest that the long-term implant does not cause damage to the nerve to the degree that would affect vital bodily functions. Importantly, the fact that these preserved reflexes are mediated by large, myelinated fibers further supports the viability of the nerve after chronic cuffing as large fibers are the most sensitive to chronic compression injury (*Dahlin et al., 1989*).

The potential wide therapeutic applicability of VNS lies in part in its ability to modulate inflammation, a process fundamental to the pathogenesis of many diseases (*Couzin-Frankel, 2010*; *Furman et al., 2019*; *Slavich, 2015*; *Strowig et al., 2012*). VNS controls inflammation by activating the IR, defined as an afferent arm that responds to inflammation in the periphery via vagal sensory neurons and an efferent arm that suppresses activation of macrophages via vagal motor neurons that project to the spleen (*Tracey, 2009*; *Tracey, 2002*). This effect was first described in a model of LPS endotoxemia, wherein acute VNS suppressed TNF release (*Borovikova et al., 2000*), and has since been established as a standard functional read-out of the anti-inflammatory effects of VNS (*Caravaca et al., 2019*). For these reasons, we sought to determine whether stimulation through long-term implants could reproduce that effect. We found that delivering three doses of stimulation to the animals over 2 days using 9- to 17-day-old implants significantly decreased serum TNF compared to sham-stimulated and sham surgery animals. In contrast, delivering a single stimulation dose, using older implants (about 3–6 weeks old) did not result in significant reduction in TNF. However, we observed a considerable decrease in serum TNF (>~40% compared to mean TNF in sham stimulation controls) in four out of seven animals with functional cuffs, as demonstrated by VNS-elicited HR responses (*Figure 5E*). The lack of an anti-inflammatory response in the single-dose group could have several explanations. Chronic cuffing of the VN might suppress efferent signaling that could reduce the anti-inflammatory actions of VNS (*Somann et al., 2018*) due to loss of nerve fibers smaller than what we could detect histologically (*Nitz et al., 1986*; *Szabo and Sharkey, 1993*). Alternatively, some animals in the single-dose group could be non-responders, a common occurrence in these studies (*Caravaca et al., 2019*; *Huffman et al., 2019*). Our data indicate that a HR response does not necessarily mean engagement of the IR, consistent with previous findings in acute studies (*Huffman et al., 2019*; *Huston et al., 2007*). Anatomically, vagal cholinergic fibers signaling to the spleen via abdominal sympathetic ganglia to suppress inflammation (*Kressel et al., 2020*) may lie in distinct parts of the nerve from those innervating the heart to cause drops in HR (*Rajendran et al., 2019*). It is unknown if the IR is activated by engagement of efferent fibers, or additionally by afferent fibers, some of which are activated by lower VNS intensities. Recent reports implicate both afferent and efferent pathways (*Kressel et al., 2020*; *Murray et al., 2021*; *Olofsson et al., 2015*) in the anti-inflammatory actions of VNS, and in a study by *Huffman et al., 2019*, TNF was suppressed using 90% of HRT, without reduction in HR. Moreover, stimulating at or above HRT could activate compensatory reflexes that might alter the anti-inflammatory effect of VNS, as shown in one report where stimulation at higher frequencies and amplitudes resulted in an increase in TNF in healthy animals (*Tsaava et al., 2020*). We suggest using HRT as a practical method to estimate stimulation dose required to achieve consistent efferent, parasympathetic fiber recruitment in an implant, thereby reducing variability related to the nerve anatomy itself and the nerve-electrode interface. Stimulus intensities required to engage specific circuits and organs could then be determined experimentally relative to HRT (*Huffman et al., 2019*). A fixed set of parameters could still be used in recovered animals beyond the first week when the procedure is optimized, and variability minimized, as we show in the endotoxemia experiment with multiple doses (*Figure 5*). In this experiment, animals received VNS with 250 µs PW, compared with the 500 µs PW used in other validation cohorts. This shorter PW could explain why 1 mA did not usually result in HR changes (*Figure 5C*), as HRT is generally greater for shorter PWs (*Figure 3C*). Future studies could focus on empirically determining reliable stimulation doses to chronically engage the anti-inflammatory pathway.

Therefore, we have developed a surgical approach to implant micro-cuff electrodes onto the mouse cervical VN to deliver chronic electrical stimulation. Our findings demonstrate a stable and robust interface with the nerve capable of producing stimulus-evoked responses for at least 4 weeks. We also provide a method to standardized stimulation doses between animals and across time. Given the wide range of illnesses studied in the mouse, this implant methodology provides a previously unavailable tool for screening of long-term VNS in disease models.

## Acknowledgements

The authors thank Michael Gerber, Stefanos Zafeiropoulos, Jacquelyn Tomaio, Khaled Qanud, Shubham Debnath, Todd Levy, Maria Lopez, Joanne Peragine, Loren Reith, Moon Rob, Rongchen Huang, Spencer Bowles, and Laura Goldman for their assistance in experiments, electrodes, and data

analysis. We also thank Tom Coleman, Sangeeta Chavan, Tea Tsaava, Valentin Pavlov, and Juan Yang for their thoughtful discussions. SZ was funded by Defense Advanced Research Projects (DARPA) Biological Technologies Office/Targeted Neuroplasticity Training (HR0011-17-2-0025) and a research grant from United Therapeutics Corporation. YA was funded by a research grant from Boston Scientific. CW was funded by DARPA Biological Technologies Office/Targeted Neuroplasticity Training (HR0011-17-2-0051). RCF was funded by DARPA (N66001-17-2-4010), the Brain Research through Advancing Innovative Neurotechnologies (BRAIN) Initiative (NS107616), Eunice Kennedy Shriver National Institute of Child Health and Human Development (NICHD) (HD088411), National Institute on Deafness and other Communication Disorders (NIDCD) (DC12557), and a Howard Hughes Medical Institute Faculty Scholarship. The funders had no role in study design, data collection and analysis, decision to publish, or preparation of the manuscript.

## Additional information

### Competing interests

Kevin J Tracey: holds patents broadly related to this work. He has assigned all rights to the Feinstein Institutes for Medical Research. The other authors declare that no competing interests exist.

### Funding

| Funder | Grant reference number | Author |
| --- | --- | --- |
| Defense Advanced Research Projects Agency | HR0011-17-2-0025 | Stavros Zanos |
| United Therapeutics Corporation | | Stavros Zanos |
| Boston Scientific Corporation | | Yousef Al-Abed |
| Defense Advanced Research Projects Agency | HR0011-17-2-0051 | Cristin Welle |
| Defense Advanced Research Projects Agency | N66001-17-2-4010 | Robert C Froemke |
| Eunice Kennedy Shriver National Institute of Child Health and Human Development | HD088411 | Robert C Froemke |
| Brain Research Trust | NS107616 | Robert C Froemke |
| National Institute on Deafness and Other Communication Disorders | DC12557 | Robert C Froemke |
| Howard Hughes Medical Institute | Faculty Scholarship | Robert C Froemke |

The funders had no role in study design, data collection and interpretation, or the decision to submit the work for publication.

### Author contributions

Ibrahim T Mughrabi, Conceptualization, Data curation, Formal analysis, Validation, Investigation, Visualization, Methodology, Writing - original draft, Writing - review and editing; Jordan Hickman, Data curation, Validation, Investigation, Visualization, Methodology, Writing - review and editing; Naveen Jayaprakash, Adam Abbas, Investigation, Writing - review and editing; Dane Thompson, Formal analysis, Validation, Investigation, Methodology, Writing - review and editing; Umair Ahmed, Data curation, Investigation, Methodology, Writing - review and editing; Eleni S Papadoyannis, Methodology, Writing - review and editing; Yao-Chuan Chang, Software, Formal analysis, Writing - review and editing; Timir Datta-Chaudhuri, Eric H Chang, Theodoros P Zanos, Writing - review and editing; Sunhee C Lee, Formal analysis, Methodology, Writing - review and editing; Robert C Froemke, Kevin J Tracey, Cristin Welle, Yousef Al-Abed, Conceptualization, Funding acquisition, Writing - review and editing; Stavros Zanos, Conceptualization, Data curation, Software, Formal

analysis, Supervision, Funding acquisition, Visualization, Methodology, Writing - original draft, Project administration, Writing - review and editing

### Author ORCIDs
Ibrahim T Mughrabi https://orcid.org/0000-0001-8057-6146
Yao-Chuan Chang http://orcid.org/0000-0003-0340-4652
Robert C Froemke http://orcid.org/0000-0002-1230-6811
Stavros Zanos https://orcid.org/0000-0002-3967-8164

### Ethics
Animal experimentation: All animal experiments complied with relevant ethical guidelines and were approved by the Institutional Animal Care and Use Committee (IACUC) of the Feinstein Institutes for Medical Research (protocol numbers: 2016-029, 2017-010, and 2019-010) and University of Colorado Anschutz Medical Campus (protocol number: 00238).

### Decision letter and Author response
Decision letter https://doi.org/10.7554/eLife.61270.sa1
Author response https://doi.org/10.7554/eLife.61270.sa2

## Additional files

### Supplementary files
• Transparent reporting form

### Data availability
All data generated or analyzed during this study are included in the manuscript and supporting files. Source data files have been provided for Figures 2, 3, and 4.

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
