## [Decision Letter]

**Acceptance summary:**

The vagus nerve connects the brain to internal organs to regulate physiology and inflammation. Vagus nerve stimulation (VNS) has the potential to treat disease, but its long-term use in mice has posed significant surgical and engineering challenges. In this study, four labs collaborate to develop a long-term implant protocol for VNS in mice that successfully modulates heart rate and inflammatory readouts. This tool will make possible future studies of the therapeutic potential of VNS in chronic diseases.

**Decision letter after peer review:**

Thank you for submitting your article "An implant for long-term cervical vagus nerve stimulation in mice" for consideration by *eLife*. Your article has been reviewed by 3 peer reviewers, and the evaluation has been overseen by a Reviewing Editor and Carla Rothlin as the Senior Editor. The reviewers have opted to remain anonymous.

The reviewers have discussed the reviews with one another and the Reviewing Editor has drafted this decision to help you prepare a revised submission.

Summary:

In this manuscript, Mughrabi et al. reported a technical advance of long term vagus nerve stimulation (VNS) in mice. VNS has been used in clinics for treating certain patients with epilepsy and depression and pioneered in clinical trials for a number of disorders including inflammation. Yet, VNS has not been widely used in mice for mechanistic studies largely due to technical challenges dealing with the small size. Here, the authors developed a method for chronic implantation of VNS stimulator in mice, and tested the effectiveness of the method using measurements of heart rate changes and effects on inflammation. This method is potentially useful to investigate the therapeutic potential of long-term VNS in chronic disease models in mice. While reviewers were positive about the work performed in this study including that it was carried out by multiple labs, there are major concerns about certain points and additional essential experiments are needed. These include the need for robust data related to the LPS inflammation studies and histological analysis. There were also missing details of methodologies that decrease the enthusiasm for this study.

Essential revisions:

1. At least two papers (PMID: 28628030, 32521521) have reported implants usable for the same application (long-term VNS in mice) although more extensive validation and characterization were performed in this manuscript. A comparison between those implants and the one in this manuscript needs to be discussed. As the authors stated, one technical challenge is that the vague nerve in mice is very small and fragile. However, it is unclear how the approach presented here is different from previous designs, and in particular, how mechanical damage is reduced using the reported apparatus.

2. If the paper is going to be a resource, the authors should provide detailed descriptions of the materials and construction of the electrode. Currently the details are sparse and the photos of poor resolution. It is unclear how the custom cuff was built (no details provided in the method section), what materials were used, and whether these materials are bio-compatible. Also, it is not clear whether and how the cuff electrode is appropriately insulated to prevent stimulation of surrounding muscles/nerves. In addition, the touching point between the nerve and the cuff is very easy to be damaged. With the description of the implantation procedure, it should also be made clearer as to when the cuff electrode is place on the nerve. A clear description could prevent torsion or other injury to the nerve.

3. LPS experiments: All reviewers thought the LPS experiment needed improvement. This study is under-powered and lacks a control group (saline + Sham stim). The LPS study is inconclusive due to a small number of animals. Increasing N to get conclusive data is important because this implant will be very useful to investigate the anti-inflammatory effect of long-term VNS in chronic disease models in mice. Related to this point, out of the 4 animals with bradycardia, 2 animals did not show a decrease in serum TNF. This raises a concern that using heart rate threshold may not be appropriate to deliver a consistent stimulation dose within/across animals if the goal is to get a consistent anti-inflammatory effect. It is likely that vagus efferent fibers responsible for HR decrease (innervating the sinoatrial and atrioventricular nodes) and those responsible for an anti-inflammatory effect are different populations. Those two populations might be differently affected by the implantation surgery and repetitive stimulation. In addition, performing VNS in awake animals is closer to the human situation.

4. Please confirm that 0.1mg/kg is the correct dose, this seems low to induce this amount of TNFa.

5. The histology of the vagus nerve raised questions and needs to be addressed. Here were relevant comments by reviewers.

– In Figure 4b, the vagus nerve in the cuff is quite clear, as is the carotid artery. But there are other nerve fragments and/or auto-fluorescent tissue immediately adjacent. What are these? Leads one to wonder if they only stimulated the vagus? The cervical sympathetic travels with the cervical vagus and care is needed to separate them from the carotid sheath. On the right side of Figure 4b, the "control" side, they highlight a nerve nowhere near the carotid artery. This is intact tissue, so the vagus has to be next to the carotid artery. There is a big nerve next to the right carotid that I would bet is the vagus. I think they've got it wrong. It is not clear at what level these photos are taken, is it the cervical vagus? The authors should indicate the left and right carotid in these figures.

– Figure 4. I do not see how fibrosis is determined. Is this actually collagen? Can the sections in B be stained with mason's trichrome. In "B" I am not sure that I see that the indicated regions are in fact the vagus nerve. It is hard to tell what other nerves would be present as there are few indications of the anatomical area these sections are from other than neck. Thus it Is hard to discern if this really is the vagus or not. I would have thought that the carotid artery should be visible in close proximity to the nerve bundle, this seems not to be the case and leads to uncertainty that this is the correct nerve.

– Was there any difference in histology between mice with functioning and non-functioning cuffs? As stated in Discussion, left VN without surgery in different animals would be a better control than right VN in the same animals.

6. In the data presented in Figure 2 or any of the studies where the kent scientific pulse/ox was used, Did O2 saturation decrease with the change in breathing?

7. Why didn't animals receiving awake VNS show visible changes in BR, which is in contrast to remarkable changes in BR in anesthetized animals?

8. In video 1, it is unclear when the stimulation starts or stops. As a result, it is uncertain if the mouse scratching is due to stimulation. Is this a pain/nociceptive response?

9. Figure 3 is presented in a confusing manner. In "A", I'm not sure why two mice are presented for different days post implantation and what this is showing. There is a clear effect of VNS on the heart rate and breathing (rate, and air flow), is this the minimum current for each day that was found to induce the heart rate threshold change. While I appreciate that the longer pulse widths are less susceptible to the effect of bio-encapsulation of the electrode over time, I'm not sure how one compares 100 uA at 100 us to 400 uA at 600 us. In B how is the HRT achieved without damaging the electrode as the ICIC is exceeded, or are we not understanding this graph correctly? In C there are days that seem to be missing given the legend. The supplementary figure also appears to have data points missing or obscured?

10. Success rate tops out at 75% with a skilled surgeon, and ranges between 40-60% for your average player. I'd say this is not too good.

11. It would be nice to show that the implant does not cause chronic inflammation as this would impact its usefulness as a method. The authors should measure tnfa 14 days Post implanted in cuff implanted and sham implanted mice.

12. What behavioral experiments were done, and what were the results? These are mentioned in several places (line 172, line 279 etc) but not reported.

13. The vagus nerve is critically involved in many essential body functions. Chronic implantation of the VNS stimulator may cause severe inflammation, nerve damage, and neuronal dysfunction. Therefore, it is critical to demonstrate that the chronic implantation does not alter nerve function. The chronic effect of the VNS stimulator implantation needs to be carefully monitored. For example, whether there is any change in body weight, food intake, as well as the sensitivity of diverse physiological reflexes such as the baroreflex, the Hering-Breuer reflex, and the stomach accommodation reflex.

---

## [Author Response]

Essential revisions:1. At least two papers (PMID: 28628030, 32521521) have reported implants usable for the same application (long-term VNS in mice) although more extensive validation and characterization were performed in this manuscript. A comparison between those implants and the one in this manuscript needs to be discussed. As the authors stated, one technical challenge is that the vague nerve in mice is very small and fragile. However, it is unclear how the approach presented here is different from previous designs, and in particular, how mechanical damage is reduced using the reported apparatus.

We added a paragraph in the Discussion section that addresses the two referenced papers [1, 2], in addition to a recent paper published during our preparation of the revised submission that reported using long-term VNS in mice [3]. We would like to stress, as laid out in the revised discussion, that the two referenced papers focused on long-term vagus nerve recordings; one paper reported a single stimulation event in only one animal, which is insufficient to meaningfully demonstrate efficacy for long-term VNS. Furthermore, unlike the referenced papers, which used in-house built electrodes, we developed our approach using commercially available, off-the-shelf micro-cuffs from 2 suppliers (CorTec and Micro-Leads) to ensure consistency and allow adoption by non-neuroengineering labs. We implemented multiple surgical maneuvers and device modifications to reduce mechanical damage. For example, we used micro-cuffs with 100-150 μm in diameter, which is about the same or slightly larger than the mouse VN diameter. We also opted to use coiled lead wires to reduce mechanical strain. Surgically, we tunneled the electrode under the sternocleidomastoid muscle to position the cuff on the same plane as the nerve as well as prevent the cuff from tracking laterally by providing a muscular border. Measures and manipulations aimed at facilitating long-term functionality of vagal implants were not described in the referenced papers, so we consider those papers as “proof-of-concept” efforts, with little methodological value.

Main text changes Discussion

“For these reasons, a long-term interface to deliver stimulation to the VN is needed. […] To our knowledge, this report is the first demonstration of chronic VNS in the mouse that provides longitudinal evidence of stimulus-elicited physiological responses.”

Results

“Further, tunneling deep to the sternocleidomastoid muscle helped align the cuff on the same plane as the nerve and minimized lateral tracking of the cuff by providing a muscular border. […] Potential mechanical damage was further reduced by using cuffs slightly larger (100-150 um) than the diameter of the vagus to prevent compression and by incorporating coiled wires to reduce mechanical strain.”

2. If the paper is going to be a resource, the authors should provide detailed descriptions of the materials and construction of the electrode. Currently the details are sparse and the photos of poor resolution. It is unclear how the custom cuff was built (no details provided in the method section), what materials were used, and whether these materials are bio-compatible. Also, it is not clear whether and how the cuff electrode is appropriately insulated to prevent stimulation of surrounding muscles/nerves. In addition, the touching point between the nerve and the cuff is very easy to be damaged. With the description of the implantation procedure, it should also be made clearer as to when the cuff electrode is place on the nerve. A clear description could prevent torsion or other injury to the nerve.

These experiments utilize two cuff electrodes that are commercially fabricated by MicroLeads Neuro and CorTec (see references to these in lines 80-81, and 140 in original submission). We made the following text alterations to clarify this point and added a brief description on their biocompatible construction and materials used:

1. In the electrode preparation section of the methods, we previously wrote “Bipolar platinum-iridium micro-cuff electrodes with an internal diameter of 150 μm (MicroLeads Neuro, Somerville, MA) or 100 μm (CorTec, Freiburg, Germany) were soldered to gold sockets after cutting the lead wires to a length of 2.5-3.0 cm (Figure 1A).”

We altered this to:

“150μm MicroLeads cuff electrodes (MicroLeads Neuro, Gaithersburg, MD) and 100μm CorTec micro-sling cuffs (CorTec, Freiburg, Germany) were commercially fabricated and used for cuff preparation. […] These bipolar platinum-iridium micro-cuff electrodes were soldered to gold sockets after cutting the lead wires to a length of 2.5-3.0cm (Figure 1A).”

2. In the legend of Figure 1A, we altered the text to say “Lead wires of cuff electrodes produced by MicroLeads or CorTec are cut to a length of 2.5-3.0cm and soldered to gold pins (right panel, Micro-Leads). Front facing (upper panels) and side (lower panels) view of 150 μm Micro-Leads and 100 μm CorTec cuff electrodes.”

We also used higher quality images for the micro-cuffs in panel A.

Also, the reviewer was correct in noting that the contact points on the cuff are sensitive, therefore we avoid handling that part with surgical instruments and instead manipulate the silicone shell while handling the nerve carefully to prevent stretching or torsion. We added a statement to clarify careful cuffing of the nerve to prevent damage.

Methods

“The cuff was further tunneled under the sternomastoid muscle and placed next to the VN. […] The nerve was then placed into the cuff close to the nerve’s original anatomic position and the cuff-specific closing mechanism engaged.”

We also added a statement on design features that prevent current leakage.

Results

“Also, the silicone construction of the cuff shell as well as including adequate distance between the edges of the cuff and stimulating electrodes reduced current leakage to surrounding tissues.”

3. LPS experiments: All reviewers thought the LPS experiment needed improvement. This study is under-powered and lacks a control group (saline + Sham stim). The LPS study is inconclusive due to a small number of animals. Increasing N to get conclusive data is important because this implant will be very useful to investigate the anti-inflammatory effect of long-term VNS in chronic disease models in mice. Related to this point, out of the 4 animals with bradycardia, 2 animals did not show a decrease in serum TNF. This raises a concern that using heart rate threshold may not be appropriate to deliver a consistent stimulation dose within/across animals if the goal is to get a consistent anti-inflammatory effect. It is likely that vagus efferent fibers responsible for HR decrease (innervating the sinoatrial and atrioventricular nodes) and those responsible for an anti-inflammatory effect are different populations. Those two populations might be differently affected by the implantation surgery and repetitive stimulation. In addition, performing VNS in awake animals is closer to the human situation.

We agree with the reviewer’s comments. To address these concerns, we expanded our original endotoxemia study to include more animals. Using one-time VNS delivered through ~2- to 6-week-old implants, we confirmed our previous result of lack of a statistically significant effect on TNF levels (Figure 5E). In collaboration with colleagues from Dr. Kevin Tracey’s group at Feinstein, we conducted additional experiments to test the efficacy of delivering multiple stimulation events per animal, daily, on serum TNF. We found a statistically significant reduction in TNF levels in animals that received 3 VNS events over 2 days, compared to those that received sham stimulation. We updated the manuscript with data from the multiple stimulations experiment as main Figure 5.

Regarding the use of heart rate threshold (HRT) for stimulation dosing, we understand the reviewers’ concern and would like to highlight several points. First, we agree that the fibers responsible for the anti-inflammatory effect are likely to be different from those responsible for the decrease in heart rate. Anatomically, cholinergic fibers innervating the spleen to suppress inflammation [4] may lie in distinct parts of the nerve from the cholinergic fibers innervating the heart to cause drop in HR [5]. In a study by Huffman et al. [6], the authors show suppression of TNF using 90% of HRT with no reduction in heart rate. We suggest using HRT as a practical method to estimate stimulation dose required to achieve consistent efferent, parasympathetic fiber recruitment in an implant, thereby reducing variability related to the nerve anatomy itself and the nerve-electrode interface. Stimulus intensities required to engage specific circuits and organs of interest could then be determined experimentally relative to (or in units of) HRT, as used by Huffman et al. We posit that expressing stimulation dose in units of HRT reflects the degree of nerve engagement based on the principle of ordered fiber recruitment. A fixed set of parameters could still be used in recovered animals beyond the first week when the procedure is optimized, and variability minimized. As of yet, which fibers mediate the anti-inflammatory actions of VNS is a matter of active investigation [4]. Future studies could focus on empirically determining reliable stimulation doses expressed in units of threshold to engage the anti-inflammatory pathway.

We included parts of this response in the revised discussion to highlight the points raised by the reviewer.

Methods

“…, or through a 1-cm midline incision on the dorsal neck,…”

Methods

“Alternatively, a 6-channel pedestal (F12794, P1 Technologies, Roanoke, VA), surrounded by a 2 cm circle of polypropylene mesh (PPKM404.35, Surgical Mesh, Brookfield, CT) was placed into the dorsal midline neck incision after forming a subcutaneous pocket. The incision superior and inferior to the pedestal was closed with absorbable suture and surgical clips.”

Methods

“In one set of experiments, done by the Tracey group (Feinstein Institutes), 8-week-old mice (n=12) were implanted with a left VN cuff. […] 2-6 weeks post-implantation, animals were anesthetized and received either sham stimulation or VNS at HRT intensity using 250 ms PW and 10 Hz frequency for 5 minutes. LPS was administered to mice (0.1 mg/kg, i.p.) 3 hours after stimulation.”

Results

**“**VNS using the long-term implant inhibits TNF release in endotoxemia

Acute VNS decreases serum TNF levels in acute inflammation by modulating the immune response via a neuroimmune mechanism termed the inflammatory reflex [7]. […] Mice that lacked a physiological response had TNF levels comparable to sham-stimulated controls.”

**“**Figure 5. VNS using the long-term implant inhibits TNF release in endotoxemia. (A) Mice with 9- to 17-day-old implants received 3 doses of VNS over 2 days using previously reported parameters (intensity 1 mA, PW 250 μs, frequency 30 Hz). […] Data shown as mean ± SEM. P NS (VNS with bradycardia vs sham, and VNS with vs without bradycardia) by Mann-Whitney with Bonferroni correction for multiple comparisons.”

Discussion

“For these reasons, we sought to determine whether stimulation through long-term implants could reproduce that effect. […] Future studies could focus on empirically determining reliable stimulation doses to chronically engage the anti-inflammatory pathway.”

4. Please confirm that 0.1mg/kg is the correct dose, this seems low to induce this amount of TNFa.

Main text changes Methods

“Lipopolysaccharide (LPS) from *E. coli* 0111:B4 (Sigma-Aldrich, St. Louis, MO) was dissolved in saline and sonicated for 30 minutes before administration. LPS doses were determined empirically to produce physiological levels of TNF as described in [8].”

5. The histology of the vagus nerve raised questions and needs to be addressed. Here were relevant comments by reviewers.– In Figure 4b, the vagus nerve in the cuff is quite clear, as is the carotid artery. But there are other nerve fragments and/or auto-fluorescent tissue immediately adjacent. What are these? Leads one to wonder if they only stimulated the vagus? The cervical sympathetic travels with the cervical vagus and care is needed to separate them from the carotid sheath. On the right side of Figure 4b, the "control" side, they highlight a nerve nowhere near the carotid artery. This is intact tissue, so the vagus has to be next to the carotid artery. There is a big nerve next to the right carotid that I would bet is the vagus. I think they've got it wrong. It is not clear at what level these photos are taken, is it the cervical vagus? The authors should indicate the left and right carotid in these figures.– Figure 4. I do not see how fibrosis is determined. Is this actually collagen? Can the sections in B be stained with mason's trichrome. In "B" I am not sure that I see that the indicated regions are in fact the vagus nerve. It is hard to tell what other nerves would be present as there are few indications of the anatomical area these sections are from other than neck. Thus it Is hard to discern if this really is the vagus or not. I would have thought that the carotid artery should be visible in close proximity to the nerve bundle, this seems not to be the case and leads to uncertainty that this is the correct nerve.– Was there any difference in histology between mice with functioning and non-functioning cuffs? As stated in Discussion, left VN without surgery in different animals would be a better control than right VN in the same animals.

a. We agree with the reviewer that cross-sections through the neck might introduce some doubt when trying to identify micro-structures like the vagus nerve. We did follow a systematic way in identifying the vagus using size (~100 μm) and position relative to the trachea and esophagus. However, the fact that the neck of the animal was over-extended during collection of the neck block, these relationships might slightly vary. The reason why the fluorescent structure next to the right carotid in Figure 4B (original submission) was not identified as the vagus, is its relatively large diameter and position next to the trachea. Also, the right carotid was not reliably identified in samples shown in the supplemental figure, so we could not rely on it to identify the vagus nerve. We also agree that the low power figures of mouse whole neck dissection stained with anti-neurofilament antibody shown in Figure 4B and Figure 4 supplement 1 (original submission) display varying amounts of scattered irregular positive profiles in addition to the vagus nerves. This is typical of non-specific fluorescence in mixed tissues containing vessels, fat, cartilage, muscle, and collagenous tissue. Tissue cutting artifacts contributed to the non-specific fluorescence as well. We have made sure to distinguish these non-neural tissues from the vagus nerve by ascertaining the characteristic tissue morphology, namely, regularly spaced axonal profiles clustered within a tight perineurium, as shown in higher power figures of each of the nerves.

For a more definitive approach, in our revised submission, we performed additional experiments by explanting the cuffs and isolating the enclosed vagus nerves. This time instead of taking a cross-section through the whole neck, we extracted the cuff with the intact nerve and performed histology analysis at three levels whenever possible. These confirm the identity of the tissue containing a single nerve that is cuffed. Related to this point, care was taken to isolate the vagus nerve during surgery under high magnification so that a single nerve trunk is placed in the cuff. We have also included images of explanted left vagus from naïve animals to use as controls instead of the right vagus. We have included the new images of the explanted nerve into the main Figure 6 and moved the neck cross-sections to Figure 6 supplement 1. We also removed the neck cross sections that do not clearly show the upper margin of the cuff and excluded the right vagus nerve images.

b. The reviewer is correct that the gold standard for assessing fibrosis would be performing Masson Trichrome on paraffin sections. In our revised submission, we performed H&E staining on the new isolated nerve specimens as well as Masson Trichrome. These new images are incorporated in the new main Figure 6.

c. We have not observed gross histological differences between functional and non-functional cuffs in a limited number of non-functional implants.

Main text changes Methods

“Mice with long-term implants of at least 2 weeks old (n=4) or naïve controls were anesthetized, and the implant site carefully exposed to locate and isolate the nerve relative to anatomical landmarks. […] In neck block sections, the left vagus nerve was identified either within the tissues covering the upper margin of the cuff or in the most anterior part of the neck adjacent to the cuff.”

Results

“Long-term implantation is associated with preservation of the cuffed nerve

[line 335]… To determine the impact of these processes in our long-term implants, we collected cuffed and non-cuffed left VNs from implanted mice at ~2-6 weeks post-implantation for gross and histological analysis; naïve mice were used as controls. […] In another group of animals (n=6) in which we examined cross-sections of whole neck blocks just above the cuff margin 6 weeks post-implantation, histological analysis revealed similar preserved nerves. (Figure 6—figure supplement 1C).”

6. In the data presented in Figure 2 or any of the studies where the kent scientific pulse/ox was used, Did O2 saturation decrease with the change in breathing?

Decreases in O2 saturation have been reported in rats in response to VNS and used to measure target engagement in other animal model systems including rats (sources). However, we did not find O2 saturation changes to be a reliable measurement of vagus nerve engagement. O2 saturation had high variability during baseline measurements and did not demonstrate a corresponding reduction with VNS-induced bradypnea or apnea. In Author response image 1 are data collected from experiments in rats showing SpO2 values for different breathing rates during VNS. Our data suggest O2 saturation is relatively stable despite fluctuations in BR during VNS. Additionally, in some of the testing conditions, the mice were anesthetized with isoflurane delivered with oxygen. This oxygen supplementation may mask alterations in O2 saturation that may normally accompany breathing rate fluctuations. For these reasons, we decided to not use O2 saturation as a physiologic readout of VNS.

7. Why didn't animals receiving awake VNS show visible changes in BR, which is in contrast to remarkable changes in BR in anesthetized animals?

In the original text, we report in line 242-243 “Animals receiving awake VNS did not show any signs of distress or visible changes in BR.” We did not notice any visible changes to breathing rate during awake VNS. We postulate that changes in breathing rate in awake animals may be more difficult to visually inspect due to movement, higher baseline breathing rates compared to anesthetized mice, and compensatory mechanisms. There may still exist small alterations to breathing rate that went unnoticed visually. Human patients with VNS for epilepsy do not experience any change in tidal volume or respiratory rate during wakefulness [17], yet can experience decreases in respiratory airflow when VNS is activated during sleep [18]. While the precise circuit mechanism that prevents respiration changes during wakefulness is not known, the animals may experience a similar phenomenon.

Main text changes:

We included this response in the discussion

“We, however, did not observe visible changes to BR during awake VNS. Even though there may still exist small alterations in BR that went unnoticed visually, changes in BR in awake animals may be more difficult to visually inspect due to movement, higher baseline BR compared to anesthetized mice, and compensatory mechanisms that are suppressed during anesthesia. Patients receiving VNS for epilepsy do not show changes in tidal volume or BR during wakefulness [17], yet exhibit decreases in airflow when VNS is activated during sleep [18].”

8. In video 1, it is unclear when the stimulation starts or stops. As a result, it is uncertain if the mouse scratching is due to stimulation. Is this a pain/nociceptive response?

We added on-screen text corresponding to when stimulation starts and stops. The mouse scratching was spontaneous and not related to stimulation as shown in the revised video.

9. Figure 3 is presented in a confusing manner. In "A", I'm not sure why two mice are presented for different days post implantation and what this is showing. There is a clear effect of VNS on the heart rate and breathing (rate, and air flow), is this the minimum current for each day that was found to induce the heart rate threshold change. While I appreciate that the longer pulse widths are less susceptible to the effect of bio-encapsulation of the electrode over time, I'm not sure how one compares 100 uA at 100 us to 400 uA at 600 us. In B how is the HRT achieved without damaging the electrode as the ICIC is exceeded, or are we not understanding this graph correctly? In C there are days that seem to be missing given the legend. The supplementary figure also appears to have data points missing or obscured?

The traces presented in Figure 3A show examples of heart rate (HR) and breathing rate (BR) responses across time in 2 different animals. The physiological traces provide a visual of the pattern of change in HR and BR as the implant ages. The stimulation parameters included on the traces represent the heart rate threshold (HRT) and are presented to highlight the change in parameters required to produce similar responses across time and animals and are not included for comparison. We edited the description of Figure 3A to emphasize these points and avoid confusion.

Figure 3

“(A) Examples of physiological responses to VNS in 2 mice, showing changes in breathing rate (green trace) and heart rate (red trace) elicited by a train of VNS with HR threshold intensity, as determined on that day of testing (yellow trace, with stimulus parameters shown). Heart rate threshold (HRT) is defined as the stimulation intensity required to produce an ~5-15% decrease in HR.”

The reviewer is correct in noting that in some animals in Figure 3B HRT exceeded the ICIC. We do mention in the discussion that ICIC was not exceeded in most animals, but not all. This likely did not cause any damage to the electrode as the stimulation events were brief (10 seconds) and limited in number. In experiments where VNS is delivered more frequently for longer periods of time (minutes to hours daily), care should be taken not to exceed ICIC. However, this shouldn’t be a limiting factor as we observed that HRT values tend to decrease as surgical proficiency increases. This was clarified in the discussion.

Discussion

“On the few occasions when ICIC was exceeded, it likely did not cause significant damage to the electrode as the stimulation events were brief (10 seconds) and limited in number. […] In such settings care should be taken not to exceed ICIC.”

We thank the reviewer for pointing out the obscured points. To better demonstrate global trends in impedance and HRT changes (Figure 3B and D), we plotted weekly pooled data from all three cohorts, grouping multiple measurements together under a corresponding week instead of plotting each cohort individually. We also removed references to individual cohorts in Figure 3 Results section for clarity. The obscured points in Figure 3C were corrected. Additionally, we added a figure that depicts pooled daily changes of HRT over the first week and a figure plotting pre-implantation impedance vs initial HRT (Figure 3—figure supplement 1A and B).

10. Success rate tops out at 75% with a skilled surgeon, and ranges between 40-60% for your average player. I'd say this is not too good.

Success rates at 30+ days are 63% across multiple surgeons at varying experience levels (40% – 85%). We found that practice leads to improvements in success rate, as demonstrated in cohorts 1 to 3, performed by the same surgeon. This surgeon was a naïve surgeon for Cohort 1, with a success rate of 40%. This surgeon’s success rate improved in Cohort 2 (70%) and Cohort 3 (90%), over approximately 3 months. In our experience, learning curves for complex surgical procedures in mice are common and expected, yet rarely reported in publications.

There are tradeoffs for doing this procedure in mice versus larger animal models. However, mice are a widely used animal model and provide a unique opportunity for manipulation of physiology; chronic VNS in mice can be utilized with other tools that are exclusively established in mice for better causal dissection of physiology.

The variability in performance could be in part due to the fact that these experiments were performed by 3 different groups. However, individuals from all collaborating labs were able to perform the surgery successfully and demonstrate VNS efficacy in their studies.

11. It would be nice to show that the implant does not cause chronic inflammation as this would impact its usefulness as a method. The authors should measure tnfa 14 days Post implanted in cuff implanted and sham implanted mice.

Mice with long-term implants did not exhibit signs of chronic inflammation at the implant site, including swelling or decrease movement because of pain. Also, upon explanation, the implant site appeared healed with no signs of infection. To test for the presence of subclinical chronic inflammation in those mice. We measured serum TNF in implanted non-stimulated and sham implanted mice 2-3 weeks post-implantation and found that these animals had no elevated levels of TNF (as shown in Figure 4—figure supplement 1C).

We edited the original text to reflect these points in the Results section:

“The implant site appeared healed and exhibited moderate tissue growth encompassing the lead wires and cuff surfaces in animals with both functional and non-functional implants (example from 12 days post-implantation shown in Figure 6A).”

“Implanted mice do not exhibit elevated levels of serum TNF 2-3 weeks post-implantation (Figure 4—figure supplement 1C).”

12. What behavioral experiments were done, and what were the results? These are mentioned in several places (line 172, line 279 etc) but not reported.

Behavioral experiments were performed to assess the effects of VNS on learning. These results will be reported in a future manuscript. A necessary portion of these behavior experiments were developing the surgical approach and verifying physiological engagement of VNS. As such, these details are reported in this manuscript. For clarity, all references to the behavioral experiments have been removed.

13. The vagus nerve is critically involved in many essential body functions. Chronic implantation of the VNS stimulator may cause severe inflammation, nerve damage, and neuronal dysfunction. Therefore, it is critical to demonstrate that the chronic implantation does not alter nerve function. The chronic effect of the VNS stimulator implantation needs to be carefully monitored. For example, whether there is any change in body weight, food intake, as well as the sensitivity of diverse physiological reflexes such as the baroreflex, the Hering-Breuer reflex, and the stomach accommodation reflex.

We agree with the reviewer on the importance of demonstrating that the implant does not cause significant disease in animals. To that aim, as the reviewers suggested, we followed the weights and food intake of implanted animals for about 2 weeks and measured serum TNF at the end of that period and found no significant difference between implanted and sham surgery animals. We also evaluated the baroreflex in implanted mice by administering phenylephrine intravascularly and recording the reflex change in heart rate. In these experiments, we found that implanted animals had a comparable reflexive response to the increase in blood pressure compared with non-implanted mice. Specifically, implanted mice had a significant reflex decrease in heart rate and a comparable baroreflex sensitivity index, which represents the ratio of change in heart rate to the change in systolic blood pressure. Further, we evaluated the breathing responses from chronically implanted animals and found that these animals exhibited changes in breathing that include acceleration and slowing down or cessation of breathing, which represent various reflexes that engage different fiber types. We added a new figure (Figure 4) that summarizes these results and corresponding text in relevant sections as outlined below. We would also like to stress that in our method we have implemented several measures to minimize inflammation and nerve damage, including performing the surgery under strict aseptic conditions, the use of bio-compatible materials, and using cuff electrodes slightly larger than the VN diameter.

Main text changes Methods

“Sham surgery animals underwent the same procedures, including nerve isolation and manipulation and subcutaneous tunneling, without the creation of a headcap. […] For awake stimulation experiments, …”

“Baroreflex assessment

Implanted and naïve mice were anesthetized and placed on a warmed surgical platform in the supine position and instrumented with ECG leads. […]Baseline values were calculated from a 10-second window immediately before the injection.”

Results

“Long-term implantation does not impact vagally-mediated reflexes

The vagus nerve modulates several vital bodily functions via reflexes, including appetite, blood pressure, and respiration [20]. […] We found that mice with long-term implants exhibit changes in breathing (Figure 4F, Figure 4—figure supplement 1D) similar to those induced in acute VNS experiments (Figure 4—figure supplement 1E).”

“Figure 4. Vagally-mediated reflexes in animals with long-term implants. (A) Weight change at first and second week post-surgery in mice subjected to sham surgery (n = 6) and in implanted mice (n=6). […]Data is presented as mean ± SEM; NS = not significant, **P* < 0.05 by Student’s t-test with Bonferroni correction for multiple comparisons.”

Discussion

“The vagus nerve maintains homeostasis by controlling many bodily functions, including appetite, respiration, and blood pressure [20]. […] Importantly, the fact that these preserved reflexes are mediated by large, myelinated fibers further supports the viability of the nerve after chronic cuffing, as large fibers are the most sensitive to chronic compression injury [31].”

References:

1. Caravaca, A.S., T. Tsaava, L. Goldman, H. Silverman, G. Riggott, S.S. Chavan, C. Bouton, K.J. Tracey, R. Desimone, E.S. Boyden, H.S. Sohal, and P.S. Olofsson, A novel flexible cuff-like microelectrode for dual purpose, acute and chronic electrical interfacing with the mouse cervical vagus nerve. J Neural Eng, 2017. 14(6): p. 066005. DOI: 10.1088/1741-2552/aa7a42.2. Falcone, J.D., T. Liu, L. Goldman, D.P. David, L. Rieth, C.E. Bouton, M. Straka, and H.S. Sohal, A novel microwire interface for small diameter peripheral nerves in a chronic, awake murine model. J Neural Eng, 2020. 17(4): p. 046003. DOI: 10.1088/1741-2552/ab9b6d.3. Ten Hove, A.S., D.J. Brinkman, A.Y.F. Li Yim, C. Verseijden, T.B.M. Hakvoort, I. Admiraal, O. Welting, P.H.P. van Hamersveld, V. Sinniger, B. Bonaz, M.D. Luyer, and W.J. de Jonge, The role of nicotinic receptors in SARS-CoV-2 receptor ACE2 expression in intestinal epithelia. Bioelectron Med, 2020. 6(1): p. 20. DOI: 10.1186/s42234-020-00057-1.4. Kressel, A.M., T. Tsaava, Y.A. Levine, E.H. Chang, M.E. Addorisio, Q. Chang, B.J. Burbach, D. Carnevale, G. Lembo, A.M. Zador, U. Andersson, V.A. Pavlov, S.S. Chavan, and K.J. Tracey, Identification of a brainstem locus that inhibits tumor necrosis factor. Proc Natl Acad Sci U S A, 2020. 117(47): p. 29803-29810. DOI: 10.1073/pnas.2008213117.5. Rajendran, P.S., R.C. Challis, C.C. Fowlkes, P. Hanna, J.D. Tompkins, M.C. Jordan, S. Hiyari, B.A. Gabris-Weber, A. Greenbaum, K.Y. Chan, B.E. Deverman, H. Munzberg, J.L. Ardell, G. Salama, V. Gradinaru, and K. Shivkumar, Identification of peripheral neural circuits that regulate heart rate using optogenetic and viral vector strategies. Nat Commun, 2019. 10(1): p. 1944. DOI: 10.1038/s41467-019-09770-1.6. Huffman, W.J., S. Subramaniyan, R.M. Rodriguiz, W.C. Wetsel, W.M. Grill, and N. Terrando, Modulation of neuroinflammation and memory dysfunction using percutaneous vagus nerve stimulation in mice. Brain Stimul, 2019. 12(1): p. 19-29. DOI: 10.1016/j.brs.2018.10.005.7. Borovikova, L.V., S. Ivanova, M. Zhang, H. Yang, G.I. Botchkina, L.R. Watkins, H. Wang, N. Abumrad, J.W. Eaton, and K.J. Tracey, Vagus nerve stimulation attenuates the systemic inflammatory response to endotoxin. Nature, 2000. 405(6785): p. 458-462.8. Caravaca, A.S., A.L. Gallina, L. Tarnawski, K.J. Tracey, V.A. Pavlov, Y.A. Levine, and P.S. Olofsson, An Effective Method for Acute Vagus Nerve Stimulation in Experimental Inflammation. Front Neurosci, 2019. 13: p. 877. DOI: 10.3389/fnins.2019.00877.9. Somann, J.P., G.O. Albors, K.V. Neihouser, K.H. Lu, Z. Liu, M.P. Ward, A. Durkes, J.P. Robinson, T.L. Powley, and P.P. Irazoqui, Chronic cuffing of cervical vagus nerve inhibits efferent fiber integrity in rat model. J Neural Eng, 2018. 15(3): p. 036018. DOI: 10.1088/1741-2552/aaa039.10. Nitz, A.J., J.J. Dobner, and D.H. Matulionis, Pneumatic tourniquet application and nerve integrity: motor function and electrophysiology. Exp Neurol, 1986. 94(2): p. 264-79. DOI: 10.1016/0014-4886(86)90101-9.11. Szabo, R.M. and N.A. Sharkey, Response of peripheral nerve to cyclic compression in a laboratory rat model. J Orthop Res, 1993. 11(6): p. 828-33. DOI: 10.1002/jor.1100110608.12. Huston, J.M., M. Gallowitsch-Puerta, M. Ochani, K. Ochani, R. Yuan, M. Rosas-Ballina, M. Ashok, R.S. Goldstein, S. Chavan, V.A. Pavlov, C.N. Metz, H. Yang, C.J. Czura, H. Wang, and K.J. Tracey, Transcutaneous vagus nerve stimulation reduces serum high mobility group box 1 levels and improves survival in murine sepsis. Crit Care Med, 2007. 35(12): p. 2762-8. DOI: 10.1097/01.CCM.0000288102.15975.BA.13. Murray, K., K.M. Rude, J. Sladek, and C. Reardon, Divergence of neuroimmune circuits activated by afferent and efferent vagal nerve stimulation in the regulation of inflammation. J Physiol, 2021. DOI: 10.1113/JP281189.14. Olofsson, P.S., Y.A. Levine, A. Caravaca, S.S. Chavan, V.A. Pavlov, M. Faltys, and K.J. Tracey, Single-pulse and unidirectional electrical activation of the cervical vagus nerve reduces tumor necrosis factor in endotoxemia. Bioelectronic Medicine, 2015. 2(1): p. 37-42.15. Tsaava, T., T. Datta-Chaudhuri, M.E. Addorisio, E.B. Masi, H.A. Silverman, J.E. Newman, G.H. Imperato, C. Bouton, K.J. Tracey, S.S. Chavan, and E.H. Chang, Specific vagus nerve stimulation parameters alter serum cytokine levels in the absence of inflammation. Bioelectron Med, 2020. 6: p. 8. DOI: 10.1186/s42234-020-00042-8.16. Ogawa, Y. and S. Kanoh, Enhancement of endotoxicity and reactivity with carbocyanine dye by sonication of lipopolysaccharide. Microbiol Immunol, 1984. 28(12): p. 1313-23. DOI: 10.1111/j.1348-0421.1984.tb00789.x.17. Banzett, R.B., A. Guz, D. Paydarfar, S.A. Shea, S.C. Schachter, and R.W. Lansing, Cardiorespiratory variables and sensation during stimulation of the left vagus in patients with epilepsy. Epilepsy Res, 1999. 35(1): p. 1-11. DOI: 10.1016/s0920-1211(98)00126-0.18. Murray, B.J., J.K. Matheson, and T.E. Scammell, Effects of vagus nerve stimulation on respiration during sleep. Neurology, 2001. 57(8): p. 1523-4. DOI: 10.1212/wnl.57.8.1523.19. Fleming, S.M., M.C. Jordan, C.K. Mulligan, E. Masliah, J.G. Holden, R.W. Millard, M.F. Chesselet, and K.P. Roos, Impaired baroreflex function in mice overexpressing α-synuclein. Front Neurol, 2013. 4: p. 103. DOI: 10.3389/fneur.2013.00103.20. Paintal, A.S., Vagal sensory receptors and their reflex effects. Physiol Rev, 1973. 53(1): p. 159-227. DOI: 10.1152/physrev.1973.53.1.159.21. Bachmanov, A.A., D.R. Reed, G.K. Beauchamp, and M.G. Tordoff, Food intake, water intake, and drinking spout side preference of 28 mouse strains. Behav Genet, 2002. 32(6): p. 435-43. DOI: 10.1023/a:1020884312053.22. Chang, R.B., D.E. Strochlic, E.K. Williams, B.D. Umans, and S.D. Liberles, Vagal Sensory Neuron Subtypes that Differentially Control Breathing. Cell, 2015. 161(3): p. 622-633. DOI: 10.1016/j.cell.2015.03.022.23. Kral, J.G. and L. Gortz, Truncal vagotomy in morbid obesity. Int J Obes, 1981. 5(4): p. 431-5.24. Dezfuli, G., R.A. Gillis, J.E. Tatge, K.R. Duncan, K.L. Dretchen, P.G. Jackson, J.G. Verbalis, and N. Sahibzada, Subdiaphragmatic Vagotomy With Pyloroplasty Ameliorates the Obesity Caused by Genetic Deletion of the Melanocortin 4 Receptor in the Mouse. Front Neurosci, 2018. 12: p. 104. DOI: 10.3389/fnins.2018.00104.25. Khound, R., J. Taher, C. Baker, K. Adeli, and Q. Su, GLP-1 Elicits an Intrinsic Gut-Liver Metabolic Signal to Ameliorate Diet-Induced VLDL Overproduction and Insulin Resistance. Arterioscler Thromb Vasc Biol, 2017. 37(12): p. 2252-2259. DOI: 10.1161/ATVBAHA.117.310251.26. Glick, G. and E. Braunwald, Relative Roles of the Sympathetic and Parasympathetic Nervous Systems in the Reflex Control of Heart Rate. Circ Res, 1965. 16: p. 363-75. DOI: 10.1161/01.res.16.4.363.27. Carr, M.J. and B.J. Undem, Bronchopulmonary afferent nerves. Respirology, 2003. 8(3): p. 291-301. DOI: 10.1046/j.1440-1843.2003.00473.x.28. Schelegle, E.S. and J.F. Green, An overview of the anatomy and physiology of slowly adapting pulmonary stretch receptors. Respir Physiol, 2001. 125(1-2): p. 17-31. DOI: 10.1016/s0034-5687(00)00202-4.29. Siniaia, M.S., D.L. Young, and C.S. Poon, Habituation and desensitization of the Hering-Breuer reflex in rat. J Physiol, 2000. 523 Pt 2: p. 479-91. DOI: 10.1111/j.1469-7793.2000.t01-1-00479.x.30. Yu, J., Deflation-activated receptors, not classical inflation-activated receptors, mediate the Hering-Breuer deflation reflex. J Appl Physiol (1985), 2016. 121(5): p. 1041-1046. DOI: 10.1152/japplphysiol.00903.2015.31. Dahlin, L.B., B.C. Shyu, N. Danielsen, and S.A. Andersson, Effects of nerve compression or ischaemia on conduction properties of myelinated and non-myelinated nerve fibres. An experimental study in the rabbit common peroneal nerve. Acta Physiol Scand, 1989. 136(1): p. 97-105. DOI: 10.1111/j.1748-1716.1989.tb08634.x.